# Observationally constrained regional variations of shortwave absorption by iron oxides emphasize the cooling effect of dust

**Vincenzo Obiso**[1,2], **María Gonçalves Ageitos**[2,3], **Carlos Pérez García-Pando**[2,4], **Jan P. Perlwitz**[1,5], **Gregory L. Schuster**[6], **Susanne E. Bauer**[1], **Claudia Di Biagio**[7], **Paola Formenti**[7], **Kostas Tsigaridis**[8,1], and **Ron L. Miller**[1]

[1]NASA Goddard Institute for Space Studies, New York, NY, USA
[2]Barcelona Supercomputing Center, Barcelona, Spain
[3]Universitat Politècnica de Catalunya, Barcelona, Spain
[4]ICREA, Catalan Institution for Research and Advanced Studies, Barcelona, Spain
[5]Climate, Aerosol, and Pollution Research, LLC, Bronx, NY, USA
[6]NASA Langley Research Center, Hampton, VA, USA
[7]Université Paris Cité and Université Paris Est Creteil, CNRS, LISA, 75013 Paris, France
[8]Center for Climate Systems Research, Columbia University, New York, NY, USA

**Correspondence:** Vincenzo Obiso (obisovincenzo@gmail.com)

**Abstract.** TS1 The composition of soil dust aerosols derives from the mineral abundances in the parent soils that vary across dust source regions. Nonetheless, Earth system models (ESMs) have traditionally represented mineral dust as a globally homogeneous species. The growing interest in modeling dust mineralogy, facilitated by the recognized sensitivity of the dust climate impacts to composition, has motivated state-of-the-art ESMs to incorporate the mineral speciation of dust along with its effect upon the dust direct radiative effect (DRE). In this work, we enable the NASA Goddard Institute for Space Studies ModelE2.1 to calculate the shortwave (SW) DRE accounting for the regionally varying soil mineralogy. Mineral–radiation interaction at solar wavelengths is calculated according to two alternative coupling schemes: (1) external mixing of three mineral components that are optically distinguished, one of which contains embedded iron oxides; (2) a single internal mixture of all dust minerals with a dynamic fraction of iron oxides that varies regionally and temporally. We link dust absorption to the fractional mass of iron oxides based on recent chamber measurements using natural dust aerosol samples. We show that coupled mineralogy overall enhances the scattering by dust, and thus the global cooling, compared to our control run with globally uniform composition. According to the external mixing scheme, the SW DRE at the top of atmosphere (TOA) changes from $-0.25$ to $-0.30\,\mathrm{W\,m^{-2}}$, corresponding to a change in the net DRE, including the longwave effect, from $-0.08$ to $-0.12\,\mathrm{W\,m^{-2}}$. The cooling increase is accentuated when the internal mixing scheme is configured: the SW DRE at the TOA becomes $-0.34\,\mathrm{W\,m^{-2}}$ with a net DRE of $-0.15\,\mathrm{W\,m^{-2}}$. The varying composition modifies the regional distribution of single scattering albedo (SSA), whose variations in specific regions can be remarkable (above 0.03) and significantly modify the regional SW DRE. Evaluation against the AErosol RObotic NETwork (AERONET) shows that explicit representation of soil mineralogy and its regional variations reduces the low bias of model dust SSA while improving the range of variability across stations and calendar months. Despite these improvements, the moderate spatiotemporal correlation with AERONET reveals remaining modeling challenges and the need for more accurate measurements of mineral fractions in soils.

## 1 Introduction

Dust aerosols are wind-blown soil particles that alter the atmospheric energy and hydrologic cycles by perturbing solar and thermal radiative fluxes (Miller et al., 2014). Estimates of the dust direct radiative effect (DRE) are still largely uncertain, mainly due to poorly constrained dust absorption (Samset et al., 2018). The single scattering albedo (SSA) indicates the fraction of radiative energy extinguished by aerosols through scattering with respect to total extinction, due to both scattering and absorption, and therefore inversely measures the intrinsic aerosol absorptive power. The SSA is a function of the particle size distribution (PSD), complex refractive index (CRI) and shape (Hansen and Travis, 1974; Mishchenko et al., 2002). Despite substantial regional variations in soil composition (Claquin et al., 1999; Journet et al., 2014), nearly all Earth system models (ESMs) assume a globally homogeneous composition for dust aerosols, thus attributing only to PSD the spatiotemporal variations of dust SSA while neglecting the contribution of the CRI that varies according to the relative abundance of different minerals. In particular, iron oxides like hematite and goethite strongly absorb radiation at ultraviolet (UV) and visible (VIS) wavelengths (Di Biagio et al., 2019; Moosmüller et al., 2012), which largely impacts the imaginary part of the CRI (IRI). In addition to underestimating the SSA variations in space and time, this neglect of composition variations has contributed to a range of assessments of the dust DRE, as climate models usually employ dust CRIs derived from measurements in specific regions that are assumed to be globally representative (Miller et al., 2006; Colarco et al., 2014). Two different absorption regimes for dust produced a DRE of opposite sign at the top of atmosphere (TOA) in western Africa, with consequent hydrologic and thermodynamic effects (Miller et al., 2004; Strong et al., 2015). This indeterminacy has consequences for understanding the dust effect upon the present-day climate along with past climates like the African humid period (Miller et al., 2014; Pausata et al., 2016). One path forward is to explicitly represent the contribution of regionally varying soil composition in the calculation of dust optical properties and the DRE (Scanza et al., 2015; Li et al., 2021).

Sokolik and Toon (1999) described an early calculation of the dust DRE based on mineral composition. They compiled the CRI for several climatically relevant minerals (such as quartz, phyllosilicates and hematite) and calculated the resulting CRI of a dust particle that was an amalgam of those minerals with prescribed proportions. They showed that absorption at solar wavelengths is dominated by hematite, an iron oxide that is often present in trace amounts accreted within other minerals. This conclusion was corroborated by Balkanski et al. (2007), who calculated that dust particles containing 1.5 % of hematite by volume in an internal mixture, including quartz, calcite and phyllosilicates, were in best agreement with retrievals from the AErosol RObotic NETwork (AERONET; Holben et al., 1998). Both studies derived the CRI of the amalgam from the CRIs of the individual minerals through a mixing rule. Commonly used rules, such as the volume-weighted mean (VM), Maxwell Garnett (MG) and Bruggeman (BG), are in general characterized by limited validity conditions that may challenge their application (Liu and Daum, 2008; Markel, 2016). The use of any single mixing rule is only approximate given the variety and complexity of the observed dust particle morphology (Scheuvens and Kandler, 2014). Moreover, estimates of hematite IRI at solar wavelengths that can be found in the literature vary widely (Zhang et al., 2015), which represents a strong uncertainty in calculating the IRI of mineral mixtures containing hematite by means of a mixing rule.

However, measurements suggest that the derivation of the particle IRI in terms of its constituent minerals can be simplified in some wavelength ranges. Moosmüller et al. (2012) showed that dust SSA at two solar wavelengths decreased in proportion to the fractional mass of iron within a collection of aerosolized soil samples. Di Biagio et al. (2019, hereafter DB19) revisited this relation by aerosolizing 19 natural soil samples collected from dust source regions worldwide and related retrievals of dust IRI at UV, VIS and near-infrared (NIR) wavelengths to the abundance of iron oxides in the dust aerosol samples. These experimental results suggest an innovative method for calculating the dust IRI at solar wavelengths as a function of the mass fraction of iron oxides that would circumvent the need for a theoretical mixing rule and an assumed IRI for iron oxides. The approach would complement the method by Scanza et al. (2015, hereafter SZ15) and Li et al. (2021), who followed Sokolik and Toon (1999) and Balkanski et al. (2007) by invoking an explicit mixing rule.

In this work, we update the NASA Goddard Institute for Space Studies (GISS) ModelE2.1 (Kelley et al., 2020) to calculate dust optical properties in the UV–VIS band, along with the DRE in the shortwave (SW) spectral region, accounting for the spatially and temporally varying mineral composition of dust aerosols that is calculated by the model (Perlwitz et al., 2015a). Our approach is based on the empirical relationships between the fractional mass of iron oxides and the dust IRI retrieved by DB19. We implement two schemes for assigning minerals to radiatively active dust types, allowing us to test the sensitivity of our results to the mixing state of minerals, which is otherwise constrained by few measurements. We assess the effect of varying composition upon the model calculation of dust optical depth (DOD), SSA and DRE. We also evaluate the model DOD and SSA against observations and retrievals from AERONET. This evaluation is challenged by the presence of other aerosol species that are also detected by AERONET. In particular, sea salt (present

mainly in coastal areas) may contribute to the coarse mode, and above all black carbon (BC) and brown carbon (BrC) affect the absorption by the aerosol mixture, even in small proportions. Hence, we filter AERONET scenes to identify hourly measurements where dust is the dominant component by applying multiple conditions based on both the size and absorption features of different aerosol species at solar wavelengths. Our approaches aim to minimize uncertainties in the modeling of dust optical properties for mineral mixtures as well as in the comparison with AERONET dust-filtered measurements.

In Sect. 2, after a description of ModelE2.1 (Sect. 2.1), we present our methodology for coupling radiation calculations to dust mineral composition (Sect. 2.2); we then illustrate our filtering technique to select dusty scenes from AERONET (Sect. 2.3). In Sect. 3, we present the effect of composition variations upon model dust optical properties (DOD and SSA) and DRE (Sect. 3.1), together with the comparison between the model and AERONET (Sect. 3.2). Finally, in Sect. 4, we discuss the remaining sources of uncertainty in our modeling method and comparison with observations.

## 2 Methods

### 2.1 Atmospheric model

The NASA GISS ModelE is a modular model that can simulate many components of the Earth system and their interaction through energy and mass fluxes, including the atmosphere and ocean circulation, atmospheric chemistry and aerosols (Schmidt et al., 2014; Kelley et al., 2020). The ModelE2.1 version is the first GISS contribution to the Coupled Model Intercomparison Project Phase 6 (CMIP6) (Kelley et al., 2020; Miller et al., 2021; Nazarenko et al., 2022). In this work, we use an upgraded version of ModelE2.1, including minor bug fixes and the new coupling between minerals and radiation (Sect. 2.2). We run global simulations with the atmospheric model (i.e., with ocean surface temperature and sea ice prescribed from observations), setting a horizontal resolution of 2.0° latitude by 2.5° longitude and 40 vertical layers extending to 0.1 hPa. Interactive dust minerals are calculated using the One-Moment Aerosol (OMA) module, a mass-based scheme for externally mixed constituents that can simulate a full set of aerosols including sulfate, nitrate, ammonium, black carbon, organic carbon, secondary organic aerosol, sea salt and dust (Bauer et al., 2020). In our experimental setup, ozone and aerosols other than dust are prescribed from simulations of the CMIP6 historical period (1850–2014) with the ModelE2.1 version of the OMA module (Bauer et al., 2020).

The general calculation of dust emission, transport and deposition in ModelE2.1 is described in Miller et al. (2006). Mineral dust is emitted when the model wind speed exceeds a globally uniform threshold of $8\,\mathrm{m\,s^{-1}}$ for dry soils that in-

creases with the local soil wetness. Dust sources correspond to regions with sparse vegetation and an abundance of easily erodible soil particles identified by topographic depressions (Ginoux et al., 2001). The global magnitude of dust emission is uncertain due to a scarcity of direct observations and the limited spatial and temporal resolution of the model wind speed that has a nonlinear influence upon emission. Typically, models adjust the relation between grid-box wind and emitted mass using a globally invariant calibration factor so that the model is in optimal agreement with a range of observations (e.g., Cakmur et al., 2006). Here, our calibrated emission yields a global model DOD near 0.02 that is within the observationally constrained range estimated by Ridley et al. (2016) (Sect. 3.1.1). Our global calibration factor is identical in all our experiments, so that differences in DOD and mass load among the experiments are the result of a varying radiative feedback upon the surface wind speed and emission (e.g., Miller et al., 2004; Pérez et al., 2006; Heinold et al., 2007; Ahn et al., 2007).

Dust particles are emitted and transported within one clay and four silt size bins ranging from 0.1 to 32 μm in diameter (Table B2). To allow a more precise representation of the dust–radiation interaction at UV–VIS wavelengths, the clay bin (from 0.1 to 2 μm in diameter) is split into four sub-bins for radiation calculations using constant proportions. In ModelE2.1, aerosol removal occurs through dry processes (i.e., the combined effect of gravitational settling and turbulent surface deposition) along with wet deposition. The latter is the result of below-cloud scavenging by impaction that depends upon aerosol size but is otherwise identical for all aerosol species (Koch et al., 1999). In addition, soluble aerosols are removed through nucleation of droplets within clouds. Dust particles are largely insoluble at emission, but during atmospheric transport their solubility increases through the formation of sulfate or nitrate coatings on the particle surface (e.g., Usher et al., 2003). Bauer and Koch (2005) found that a constant solubility of 0.5 yielded a similar global dust load compared to a model version where the formation of sulfate coatings on dust particles was explicitly calculated. Relying on these results, we also assume a globally uniform dust solubility of 0.5; however, we do not account for the hygroscopic growth of the dust particles in the radiation calculations.

Tracers for individual minerals were implemented by Perlwitz et al. (2015a). The mineral version of ModelE2.1 transports the eight minerals listed in Table 1. Hematite, as prescribed by Claquin et al. (1999), is interpreted as a proxy for a broader class of iron oxides that includes goethite. Most iron oxides are observed as trace abundances accreted within other minerals (Formenti et al., 2008). ModelE2.1 represents these accretions as an internal mixture of iron oxides with each of the remaining seven minerals (the latter referred to as "host" minerals), creating seven additional transported dust types for a total of 15 (Perlwitz et al., 2015a). We refer to each of the seven host minerals accreted with iron oxides

**Table 1.** Mass densities of transported minerals ($\rho_{\min}$) as prescribed in ModelE2.1 (Perlwitz et al., 2015a). For iron oxide tracers in pure crystalline form, the arithmetic mean between hematite ($5.260\,\mathrm{g\,cm^{-3}}$) and goethite ($4.280\,\mathrm{g\,cm^{-3}}$) is used. The density of each mineral accreted with iron oxides is calculated from the densities of the single minerals according to their fraction in the particle.

| Mineral | $\rho_{\min}$ (g cm$^{-3}$) |
|---|---|
| Illite | 2.795 |
| Kaolinite | 2.630 |
| Smectite | 2.350 |
| Calcite | 2.710 |
| Quartz | 2.655 |
| Feldspar | 2.680 |
| Gypsum | 2.312 |
| Iron oxides | 4.770 |

(the latter contributing a fixed 5 % of the particle mass) as "accretion". The accretion of a small fraction of iron oxides within another mineral allows them to travel farther, as in pure form they would settle more rapidly due to their larger density (Table 1).

The total emitted dust flux is partitioned among the prescribed size-distributed mass fractions of individual minerals that vary with the local properties of the parent soil but are constant in time. We use the Mean Mineralogical Table (MMT) of Claquin et al. (1999) which provides mineral fractions as a function of the local soil type for clay- and silt-sized particles. A global atlas of soil texture gives the fractional soil mass in each of these two size categories. Additionally, since both the MMT and soil texture data refer to disturbed soils, where soil aggregates were broken by wet sieving during measurement, we reconstruct the dust aggregates that were potentially emitted from the undisturbed soil according to brittle fragmentation theory (BFT; Kok, 2011). Our reconstruction, the aerosol mineral fraction (AMF) method described by Perlwitz et al. (2015a), redistributes minerals from clay to silt sizes (diameters below and above 2 μm, respectively; Table B2). The exception is quartz, which we assume is durable and resists disintegration during wet sieving. One result of this reconstruction is the presence of phyllosilicates at silt sizes, consistent with aerosol observations (Perlwitz et al., 2015b). Moreover, we combine BFT (valid for diameters up to roughly 20 μm) with measurements during a dust event at Tinfou (Morocco; Kandler et al., 2009) to prescribe the globally uniform emitted fractions of clay mass and total silt, whose diameters extend to 32 μm in our model (Table B2). Our method, therefore, also results in enhanced emitted fractions of coarse dust particles. Finally, the size-resolved concentration measurements of individual minerals from Kandler et al. (2009) are used to distribute the mineral fractions across the four silt size bins transported by ModelE2.1 (Table B2), which implies that those fractions vary with the local soil mineral composition.

## 2.2 Coupling of dust mineralogy to radiation calculations

We demonstrate the effect of spatial and temporal variations in mineral composition upon the dust optical properties (DOD, SSA) in the UV–VIS band, along with the SW DRE, by comparing two mineral experiments ("EXT" and "INT") that account for the content of iron oxides to a control run ("HOM") that instead prescribes globally homogeneous composition. The contrast between the EXT and INT experiments reveals the sensitivity of our results to two alternative mineral–radiation coupling schemes. For each of the three experiments, we calculate a climatology from a 30-year simulation (1991–2020), setting 1 year of spinup, with greenhouse gas concentrations and a prescribed atmospheric composition of the year 2000.

We calculate the CRIs used in the mineral experiments by merging the dust IRI retrievals from DB19 with the collection of CRIs of individual minerals from SZ15. DB19 measured the SW scattering and absorption spectra, volume size distribution and bulk mineral composition of 19 dust aerosol samples generated by injecting parent soil samples within a laboratory chamber. They then retrieved the CRI of the dust aerosols at seven SW wavelengths and observed an increasing relation of the retrieved IRI to the content of iron oxides in each aerosol sample (Sect. 2.2.3). Due to their experimental setup, DB19 retrieved one CRI for each sample, thus implicitly representing dust particles as identical internal mixtures of all minerals, with a uniform composition across different sizes but varying among the samples. (AERONET similarly assumes an internal mixture of constituents that is identical at all sizes and for all particles in the column.) This description is more consistent with our INT coupling scheme (except for the variation in composition with size that we assume), whereas in the EXT scheme we assume that iron oxides are internally mixed with only a (varying) fraction of host minerals (Sect. 2.2.1).

Sections 2.2.1 to 2.2.4 describe our strategy for modeling radiatively active mineral components and calculating the corresponding CRIs. In Appendix B1, we report the CRIs used in all our experiments (Table B1) and describe in detail our computation of dust optical properties in the SW spectrum, with and without accounting for varying mineral composition.

### 2.2.1 Mixing configurations for minerals

To relate mineral composition to radiation parsimoniously, we redistribute the 15 mineral tracers to a smaller number of radiatively active aerosol components, according to two alternative coupling schemes. In our first scheme (EXT), we use the mineral combinations calculated by ModelE2.1 (Sect. 2.1). For each size bin, we assemble the mass of the 15 mineral tracers into three components: (1) iron oxides by themselves ("free" or "crystalline" iron oxides), (2) the remaining seven host minerals together and (3) the seven ac-

cretions of iron oxides within each host mineral all together. We then assign a different (globally uniform) CRI to each of the three mass components by assuming a simplified optical model: the host minerals are considered a globally uniform amalgam ("host mixture"), while the accretions are modeled as an internal mixture of the host amalgam and a fixed mass fraction of iron oxides ("static accretion"). It follows that the EXT scheme does not assume a "pure" external mixing of all minerals; instead, it assumes external mixing of two internal mixtures (one of which contains embedded iron oxides) plus free iron oxides. In Fig. 1, we show a schematic cartoon of the mixing state of minerals in both our coupling schemes. Because the mass fraction of iron oxides in the static accretion is fixed at 5 % (Sect. 2.1), in the EXT scheme mineral-induced variations of dust optical properties in space and time result solely from the varying proportion of the three components. The reduction of the 15 mineral tracers to three radiative types is a simplification based on the observation that iron oxides dominate dust absorption at UV–VIS wavelengths (Sokolik and Toon, 1999; Zhang et al., 2015), while the IRIs of the seven host minerals are much smaller (Scanza et al., 2015).

In the second coupling scheme (INT), we assume that all minerals (including iron oxides) are part of a single size-resolved internal mixture. We refer to this mixture as "dynamic accretion", in contrast with the static accretion of our model-derived EXT scheme, because the particle composition in each bin and the associated CRI vary locally according to the constantly changing proportion of iron oxides at any grid box and time step (Fig. 1). While our model explicitly calculates how iron oxides are distributed into particles, there are a few observations to test our default configuration. Thus, this second scheme tests the sensitivity of the dust SW DRE to the assumed mixing state of iron oxides with other minerals.

Both the EXT and INT coupling schemes relate the CRI to mineral composition in only the UV–VIS band of the radiation module of ModelE2.1 (0.30–0.77 µm), where in contrast we assume a globally homogeneous composition in the HOM scheme (see below). Slightly more than half of the total insolation at the TOA is within this band, where recent measurements from DB19 show that most SW absorption by dust occurs (Sect. 2.2.3). In the remaining five NIR bands of the SW spectrum (covering wavelengths from 0.77 to 4 µm), we use unspeciated dust optical properties that are identical in all our experiments. We calculate the default homogeneous properties using a globally uniform CRI prescribed by smoothly joining the IRI retrieved by Sinyuk et al. (2003) at UV–VIS wavelengths with values reported by Volz (1973) at wavelengths longer than 2.5 µm, extending into the longwave (LW) bands. Similarly, for the real refractive index (RRI), values from Patterson et al. (1977) at UV–VIS wavelengths are smoothly joined to the values from Volz (1973) in the LW spectrum. In the control experiment (HOM), we use the default CRI for all the minerals, also in the UV–VIS band, thus

prescribing homogeneous composition. In this work, we do not update the calculation of dust optical properties in the LW bands and use the default properties of the model that explicitly only account for dust absorption. To represent the effect of LW scattering, the extinction is increased by 30 % (Miller et al., 2006). Moreover, in all the optical calculations conducted in this work, we assume spherical shape for dust particles.

### 2.2.2 Complex refractive index of the host mixture

In the EXT scheme, we need the CRI for the radiative component consisting of host minerals (host mixture), which we also use as the CRI for the static (EXT) or dynamic (INT) accretions in the limit of vanishing iron oxides. SZ15 showed that differences among CRIs of individual host minerals are of secondary importance compared to their contrast with the iron oxide CRI at solar wavelengths. Thus, we derive a globally uniform CRI for the host amalgam, neglecting the regional variations in the soil fractions of host minerals. We apply a mixing rule to the individual CRIs of the seven host minerals (taken from SZ15; Fig. 2) according to the mass fractions of the same minerals in 18 dust aerosol samples measured by DB19 (reported in the supplement of Di Biagio et al., 2017). (We exclude the "Taklimakan" sample of DB19 because no measurements of iron oxides were made on this sample.)

Homogenization theories such as the MG and BG methods are among the common mixing rules for calculating the CRI of particles composed of different constituents. MG assumes the particle to be composed of a homogeneous host material filled with small inclusions of contrasting composition, but its application is challenged by the requirement that the total volume fraction of the inclusions must be much lower than the host volume fraction (Markel, 2016). BG generalizes MG to a particle morphology in which the inclusions virtually occupy the entire particle volume (and the host disappears) but requires numerical methods for calculating a physical solution for mixtures of several minerals (Markel, 2016). Given these limitations, we calculate the CRI of the host mixture using the VM rule, which prescribes the real and imaginary parts of the composite CRI in proportion to the volume fractions of the constituent minerals. The VM rule can be derived as a linear approximation to the Lorentz–Lorenz mixing rule for the case of a quasi-homogeneous mixture of different components with similar refractive indices (Liu and Daum, 2008). In a first approximation, this condition applies to our host mixture that is composed of minerals with relatively similar refractive indices. While the VM rule predicts greater absorption compared to MG or BG when the inclusion is highly absorbing, this bias is less important for the amalgam of host minerals whose absorption is small compared to iron oxides.

To obtain the CRI of the host mixture, we first calculate mineral volume fractions from the mass fractions in

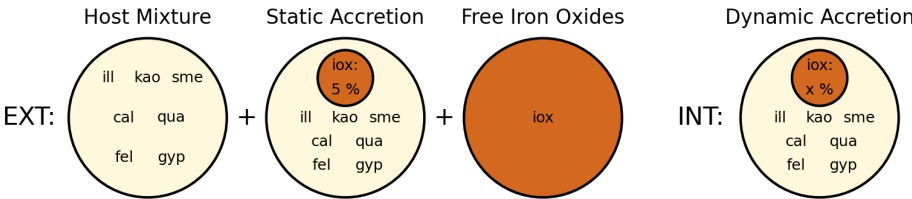

**Figure 1.** Schematic cartoon of the mixing state of minerals, assumed for radiation calculations, in the coupling schemes with external (EXT) and internal (INT) mixing configurations. The EXT scheme assumes external mixing of three mineral components, whose size-resolved proportions vary in space and time: (1) a homogeneous amalgam of non-iron oxide minerals (host mixture), (2) a two-component mixture of the host amalgam and a fixed 5 % mass fraction of iron oxides (static accretion) and (3) free iron oxides. In the INT scheme, the 15 mineral tracers are re-arranged to form a single size-resolved internal mixture of the host amalgam with a mass fraction of iron oxides that varies at each grid box and time step (indicated as $x$ %). The acronyms for minerals indicate illite ("ill"), kaolinite ("kao"), smectite ("sme"), calcite ("cal"), quartz ("qua"), feldspar ("fel"), gypsum ("gyp") and iron oxides ("iox").

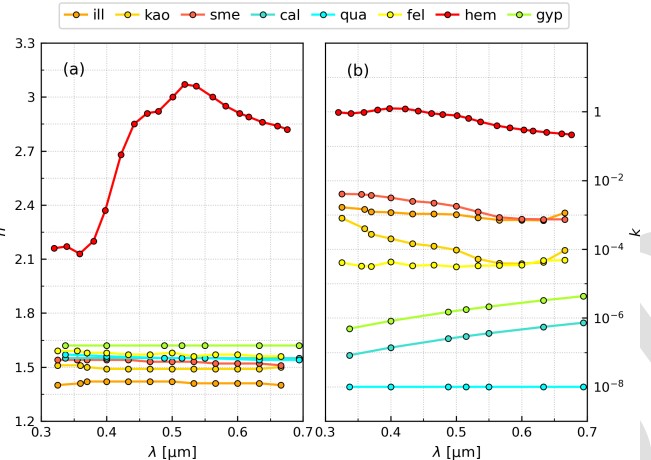

**Figure 2.** Real ($n$; **a**) and imaginary ($k$; **b**) indices of single minerals at visible wavelengths ($\lambda$) as collected by Scanza et al. (2015): illite ("ill"), kaolinite ("kao"), smectite ("sme"), calcite ("cal"), quartz ("qua"), feldspar ("fel"), hematite ("hem") and gypsum ("gyp").

the DB19 aerosol samples using the mass density of individual host minerals as prescribed in ModelE2.1 (Table 1). We then apply the VM rule to each sample and calculate the mean across all the samples. We perform this calcula-
5 tion at the seven solar wavelengths of DB19 (0.370, 0.470, 0.520, 0.590, 0.660, 0.880 and 0.950 µm) after interpolating the mineral CRIs from SZ15 at these wavelengths. Importantly, we find very good agreement between our calculated host mixture IRI and the IRI retrieved by DB19 in the sam-
10 ples with iron oxide fractions approaching zero. For example, in the Bodélé sample, whose measured mass fraction of iron oxides is only 0.7 %, DB19 retrieved IRI values of 0.0007 and 0.0004 at 0.470 and 0.590 µm, respectively. At the same wavelengths, our VM calculation for the host mixture gives
15 0.00077 and 0.00035, respectively. This agreement (within the experimental error of DB19) gives us confidence in our independent derivation of the IRI for the host mixture and

allows us to use it as the basis for the calculation of the accretion IRI.

### 2.2.3 Complex refractive index of accretions

We model our static and dynamic accretions as two-component particles consisting of the host mixture plus a small fraction of iron oxides. We calculate the IRI of these accreted particles using the IRI of the host amalgam plus a perturbation that is proportional to the mass fraction of iron 25 oxides, which can be either fixed (static accretion) or varying (dynamic accretion). This approach is suggested by the empirical relations of dust IRI to the content of iron oxides measured by DB19 and allows us to circumvent choosing an approximate theoretical mixing rule along with a hematite 30 IRI from the large range reported in the literature (e.g., Zhang et al., 2015).

At each measurement wavelength of DB19, we perform a third-order polynomial fit to their IRI retrievals versus the mass fraction of iron oxides in their samples (Fig. 3). The 35 polynomial fit is constructed to match the host mixture IRI (Sect. 2.2.2) in the limit of a vanishing iron oxide mass fraction while reducing the low bias of a linear fit at larger fractions. We define a suitable fitting function as the third-order Maclaurin expansion of an exponential function in the mass 40 fraction of iron oxides ($m_{iox}$):

$$
\begin{aligned}
k_{acc} &= k_{hos}\, e^{p_1 m_{iox}} \\
&\approx k_{hos} \left( 1 + p_1 m_{iox} + \frac{p_1^2}{2} m_{iox}^2 + \frac{p_1^3}{6} m_{iox}^3 \right),
\end{aligned} \quad (1)
$$

**TS2** where $k_{acc}$ and $k_{hos}$ indicate the IRIs of the accretion and host amalgam, respectively, and $p_1$ is the best fit parameter defining the fitting function. DB19 also reported rela-45 tionships of dust IRI with respect to the separate mass fractions of hematite, goethite or total iron (including iron from non-iron oxide minerals like phyllosilicates). We use the total iron oxide mass as the independent variable because DB19 found a more robust correlation using hematite and goethite 50 together rather than these minerals separately. As already

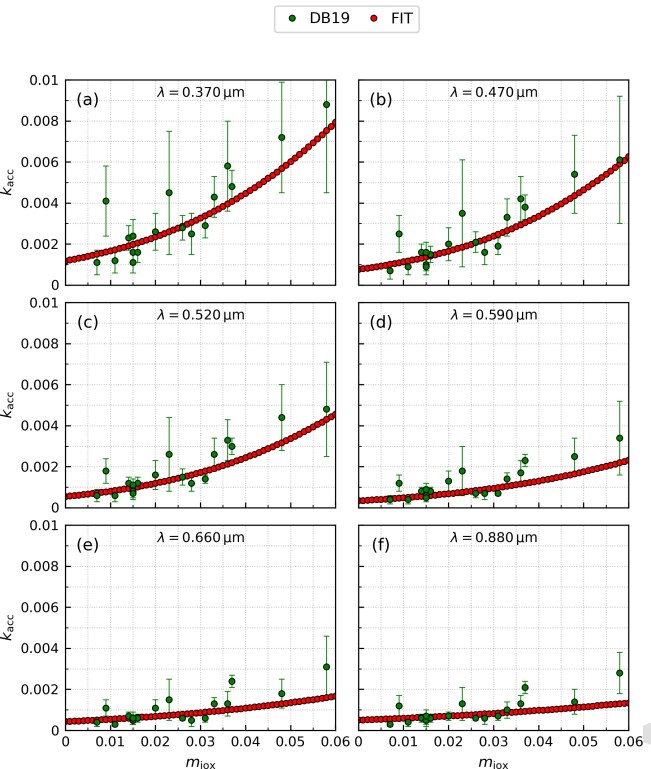

**Figure 3.** Polynomial fits (FIT) of dust imaginary indices ($k_{acc}$) retrieved by Di Biagio et al. (2019) (DB19) versus the mass fraction of iron oxides in the dust aerosol samples ($m_{iox}$), including both hematite and goethite. The fits are calculated at the seven shortwave wavelengths ($\lambda$) of DB19: 0.370 **(a)**, 0.470 **(b)**, 0.520 **(c)**, 0.590 **(d)**, 0.660 **(e)**, 0.880 **(f)** and 0.950 μm (last wavelength not shown). The intercept of the polynomial fitting function for each wavelength is set to the imaginary index of the host mixture (Sect. 2.2.2).

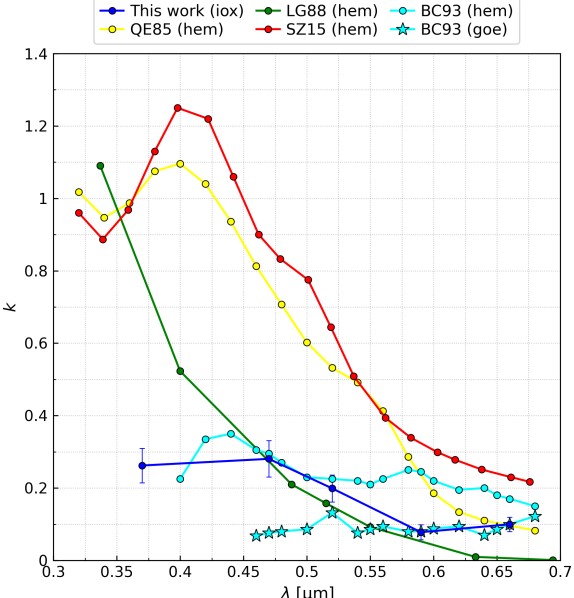

**Figure 4.** Imaginary index ($k$) of iron oxides ("iox") calculated in this work and literature values for hematite ("hem") and goethite ("goe") at visible wavelengths ($\lambda$). The references are Querry (1985) (QE85), Longtin et al. (1988) (LG88), Scanza et al. (2015) (SZ15) and Bedidi and Cervelle (1993) (BC93).

stated (Sect. 2.1), while only hematite is prescribed in the soil mineralogy of Claquin et al. (1999) and implemented in ModelE2.1 (Perlwitz et al., 2015a), we assume, as a first approximation, that hematite is a proxy for both iron oxides.

5 Perturbations of RRI due to iron oxides are less critical to our results, as dust absorption and DRE are mostly sensitive to the IRI. Moreover, given that typical fractions of accreted iron oxides are small, different mixing rules for RRI would result in a similar RRI for the accretion. For example, the ab-10 solute difference in RRI between VM and MG is smaller than 0.005 for iron oxide mass fractions below 6% (Fig. A1b). Given this insensitivity, we apply a simple two-component VM rule for the accretion RRI ($n_{acc}$), using the hematite RRI from SZ15 for iron oxides ($n_{iox}$):

$$15 \quad n_{acc} = n_{hos} + (n_{iox} - n_{hos})\, v_{iox}, \qquad (2)$$

where $n_{hos}$ indicates the RRI of the host amalgam, and $v_{iox}$ is either the static (EXT) or dynamic (INT) volume fraction of accreted iron oxides.

### 2.2.4 Complex refractive index of free iron oxides

In the EXT scheme, we also assign a CRI to the radiative 20 component comprised of crystalline or free iron oxides. As the RRI, we use the hematite value from SZ15, as already mentioned in Sect. 2.2.3. Given the large range of literature values (e.g., Zhang et al., 2015), we choose instead to derive a new IRI consistent with the measurements of DB19 through 25 an inverse calculation. This inversion requires a mixing rule to relate the retrieved IRI of composite particles to that of embedded iron oxides. Because free iron oxides make only a small contribution to the total dust extinction in the EXT scheme (Sect. 3.1.2), the specific CRI that we assign to this 30 component is nearly irrelevant to our results. Therefore, we defer our derivation of the iron oxide IRI to Appendix A. Nonetheless, in Fig. 4 we compare our calculated IRI, reflecting the presence of both hematite and goethite in the DB19 samples, to literature values for hematite and goethite. 35 Our estimate is roughly intermediate between hematite and goethite IRIs measured by Bedidi and Cervelle (1993) that, to the best of our knowledge, is the only work reporting goethite IRI at VIS wavelengths. Other estimates of the IRI for hematite alone are much higher than our values, espe- 40 cially towards short-VIS wavelengths. This discrepancy may be due to different measurement techniques and samples as well as the presence of goethite in the DB19 samples.

## 2.3 Filtering of dust events in AERONET

In order to evaluate our model calculations, we filter hourly AERONET data (Version 3; Sinyuk et al., 2020) to identify scenes where dust is the dominant aerosol and generate monthly climatologies of aerosol optical depth (AOD) and SSA (among other inversion variables) over the 2000–2020 period at the four wavelengths of the almucantar scan (0.440, 0.675, 0.870 and 1.020 μm). One of the quality assurance criteria applied to produce the Level 2.0 inversion product (Holben et al., 2006) is a threshold for AOD at 0.440 μm, $\tau_{440} > 0.4$, which ensures a relatively small error of the inversion variables but excludes the bulk of the data at many stations (Li et al., 2014). We generate two observational data sets to compare different variables: for SSA we use Level 2.0 data, whereas for DOD we apply to Level 1.5 data all of the high-quality Level 2.0 criteria except the threshold for AOD at 0.440 μm. These two data sets include a different number of stations and available months; they are indicated hereafter as AeroTAU4 ($\tau_{440} > 0.4$) and AeroTAU0 ($\tau_{440} > 0$).

Our approach for filtering dust measurements uses both the size and spectral absorption features of the common tropospheric aerosol species aiming to improve the widely used technique based only on the differences in size among species. The volume size distribution of mineral dust typically presents a pronounced coarse mode (particle diameters above ∼ 1.2 μm), especially near dust sources (Dubovik et al., 2002), whereas the fine mode is mainly populated by other species such as BC, BrC, non-absorbing organics, sulfates and nitrates. We therefore first reduce the contribution of fine species by imposing a maximum for the fine-volume fraction (FVF): $v_{\mathrm{fin}} < 0.1$. Separating dust from fine species is a fundamental first step, but a pronounced coarse mode may also indicate sea salt, especially in coastal stations (note that, in the AeroTAU4 data set, the threshold for AOD at 0.440 μm should filter out most scenes dominated by coarse sea salt, whose load does not often generate $\tau_{440} > 0.15$; Dubovik et al., 2002). Moreover, small amounts of highly absorbing carbonaceous species may contaminate the measurements, obscuring absorption by dust. To address these issues, in addition to the size filter, we use the known spectral absorption features of the different aerosol species at solar wavelengths.

Dubovik et al. (2002) showed that the aerosol mixtures observed by AERONET photometers typically exhibit decreasing or nearly constant SSA at UV–VIS wavelengths. In the case of mixtures dominated by fine anthropogenic species, negative SSA slopes may be attributable to size effects (the scattering efficiency peaks at wavelengths similar to particle size; Hansen and Travis, 1974) as well as the presence of BC which also absorbs at long-VIS wavelengths (Kirchstetter et al., 2004). In the case of pure oceanic aerosol, the nearly flat wavelength dependence of SSA is due to the substantially null absorption by sea salt in the UV–VIS–NIR spectrum. In contrast, mineral dust is the only species whose SSA increases between 0.440 and 0.675 μm (Dubovik et al., 2002), because dust aerosols are significantly composed of coarse particles and also because iron oxides are known to absorb mainly at UV and short-VIS wavelengths (Zhang et al., 2015). To further reduce contamination by non-dust species, we therefore require the SSA to increase through UV–VIS wavelengths (Ginoux et al., 2010): $\omega_{675} - \omega_{440} > 0$.

Even after this filtering, small remaining fractions of absorbing carbonaceous species may still alter the absorption capability of the mixture dominated by dust. This may happen for two reasons. First, BC and BrC particles from biomass burning and industrial or urban combustion sources have been actually found attached to dust particles (Derimian et al., 2008; Hand et al., 2010). Second, the inversion algorithm of AERONET assumes a homogeneous internal mixture of aerosols, without distinguishing between fine and coarse modes (i.e., only one CRI is retrieved for both modes): this potentially causes absorption by fine particles to be artificially shared with the coarse mode (Schuster et al., 2016). To reduce this effect, following Schuster et al. (2016), we apply our third condition, $k_{\mathrm{rir}} < 0.0042$, where $k_{\mathrm{rir}}$ is the mean IRI at red–NIR wavelengths (0.675, 0.870 and 1.020 μm). This condition directly detects the presence of BC, which is the only species absorbing at these wavelengths (Kirchstetter et al., 2004), but indirectly also filters out primary BrC that is expected to coexist with BC as a byproduct of the same combustion processes. The threshold of 0.0042 has been proposed by Schuster et al. (2016) to separate "pure" dust retrievals at Middle Eastern AERONET sites (identified by a FVF below 0.05 and a depolarization ratio above 0.2) from retrievals of "pure" carbonaceous samples at South American sites during the peak season of biomass burning: the threshold value of 0.0042 allows an overlap of only 5 % between the distributions of IRI at red–NIR wavelengths retrieved for dust and biomass burning species, with the former generally exhibiting a lower IRI. (The aerosolized soil samples analyzed by DB19 also uniformly exhibit $k_{\mathrm{rir}} < 0.0042$ at these wavelengths.) Our combination of filters aims to select measurements dominated by mineral dust, but it must be pointed out that removing all contributions by non-dust species in the AERONET observations is impossible.

For comparison to the model, we calculate climatological monthly means of DOD measurements and SSA retrievals using the AeroTAU0 and AeroTAU4 data sets, respectively. We select only stations that provide more than 80 hourly measurements per month between 2000 and 2020, for at least 1 month, after the multiple filtering. (In this work, with dust SSA and DOD from AERONET, we always refer to retrieved and measured values, respectively, in our sample of selected dusty scenes.) Our chosen threshold for the minimum number of measurements in each month represents a compromise between minimizing the uncertainty of the monthly mean while maximizing the number of stations and months in the sample. However, the threshold contributes an intrinsic uncertainty to the comparison between the model

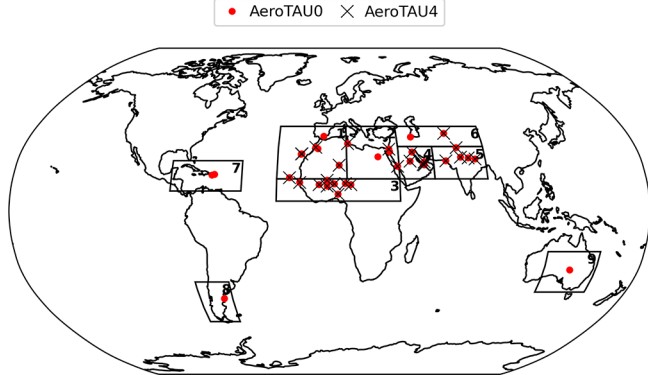

**Figure 5.** AERONET stations selected by filtering dust events located within dust regions: (1) northwestern Africa, (2) northeastern Africa, (3) Sahel, (4) Arabian Peninsula, (5) northern India, (6) central Asia, (7) Caribbean, (8) Patagonia and (9) Australia. Stations included in the AeroTAU0 data set are located within all nine regions, whereas stations in the AeroTAU4 data set are located only within the first six source regions.

monthly output and AERONET. From the monthly values at the four AERONET wavelengths, we average DOD and SSA in the UV–VIS band of ModelE2.1 (0.30–0.77 μm) by applying the same procedure used for the model properties (Appendix B2). (Note that we always calculate DOD-weighted means for both model and observed SSA.) All 38 (AeroTAU0) and 27 (AeroTAU4) stations selected through our filtering method are listed with their coordinates in Table 2 and are plotted within nine and six regions for AeroTAU0 and AeroTAU4, respectively, in Fig. 5.

## 3 Results

In the following Sect. 3.1 we show the results of our model calculations, analyzing the changes to DOD and SSA in the UV–VIS band, along with the SW DRE, due to the varying mineral composition compared to default homogeneously composed dust, and the sensitivity of these variables to the mixing configuration assumed for minerals. In Sect. 3.2, we compare the modeled dust optical properties, calculated with the two different coupling schemes, to AERONET observations in the selected stations and months.

### 3.1 Mineralogy impact upon dust optical properties and the direct radiative effect

#### 3.1.1 Evaluation of default homogeneously composed dust (HOM)

In Fig. 6, we show the annual mean DOD, SSA and all-sky DRE at the TOA and at the surface (SFC), as calculated in the control run (HOM) that assumes a globally homogeneous dust composition. The optical properties (DOD and SSA) are averaged over the UV–VIS band of ModelE2.1 (0.30–

0.77 μm), whereas the DRE is for the entire SW spectral region. Our model simulates the highest DOD (Fig. 6a) in northern Africa and the Sahel, including dust plumes downwind over the Atlantic Ocean, along with the Arabian Peninsula (DOD up to 0.91), while lower values are calculated in northern India, central Asia and Australia. In Fig. 6b, our default dust, whose IRI at UV–VIS wavelengths is from Sinyuk et al. (2003), shows SSA between $\sim 0.90$ and $\sim 0.93$ in the main dust source regions and increasing SSA downwind from the sources, indicating enhanced scattering due to the shorter lifetime of more absorbing large particles (Table 3) that are preferentially removed by gravitational settling (Miller et al., 2006). The SW DRE at the TOA, in Fig. 6c, shows a typical pattern with negative values above dark oceanic and vegetated areas, while null or slightly positive values (with peaks in specific areas) occur over bright desert surfaces. Finally, dust has a SW cooling effect everywhere at the SFC (Fig. 6d).

As reported in Table 3, in the control run, the global emission rate is $6146\,\mathrm{Tg\,yr^{-1}}$, which decreases to $4030\,\mathrm{Tg\,yr^{-1}}$ for diameters smaller than 20 μm. The latter is estimated by including all bins covering diameters below 16 μm and a fraction of the coarsest bin (16–32 μm in diameter; Table B2) calculated assuming a constant sub-bin volume size distribution on a logarithmic scale, the same we use for optical calculations (Appendix B2). In Fig. 7a, we show that roughly half of the emitted dust mass is in the coarsest bin, which results from our enhanced emission at large sizes (Sect. 2.1). Despite their prolific emission, the largest particles with diameters greater than 16 μm settle very rapidly with a lifetime of 0.43 d (Table 3), decreasing to 11 % their contribution to the bulk mass load (34 Tg; Fig. 7b and Table 3). In the coarse diameter range of 5–20 μm, our model reproduces fairly well (18 Tg) the observed mass load constrained by Adebiyi and Kok (2020): 17 Tg (10–29 Tg). This likely results from our enhanced emission at large sizes compensating for the rapid deposition of coarse particles that may be excessive compared to observations (van der Does et al., 2016).

Our mass load for diameters below 20 μm (31 Tg; Table 3) is close to the high end of the observed range constrained by Kok et al. (2017) (14–33 Tg), mostly due to their smaller mass load in the coarse diameter range of 5–20 μm (10 Tg, as reported by Adebiyi and Kok, 2020). In contrast, our global mean DOD (0.020, also for diameters below 20 μm; Table 3) equals the lowest value of the range used by Kok et al. (2017) (0.02–0.04 at 0.550 μm, as constrained by Ridley et al., 2016). Given that, compared to Kok et al. (2017) (as well as to Adebiyi and Kok, 2020), we have a similar mass load (13 Tg) for diameters below 5 μm (Fig. 3 of Adebiyi and Kok, 2020), our lower DOD may partly be due to the enhanced extinction efficiency of triaxial ellipsoid particle shapes used by Kok et al. (2017) compared to our assumed spheres.

Our global mean SW DRE at the TOA is $-0.25\,\mathrm{W\,m^{-2}}$ (Table 3). This value indicates lower cooling by dust com-

**Table 2.** Selected AERONET stations with coordinates, divided by region (Fig. 5). All the listed stations are included in the AeroTAU0 data set, while the ones also included in AeroTAU4 are marked with an "x".

| Station | Region | Latitude | Longitude | AeroTAU4 |
|---|---|---|---|---|
| Capo_Verde | Northwestern Africa | 16.73 | −22.94 | x |
| Granada | | 37.16 | −3.61 | |
| Izana | | 28.31 | −16.50 | |
| La_Laguna | | 28.48 | −16.32 | x |
| Ouarzazate | | 30.93 | −6.91 | |
| Saada | | 31.63 | −8.16 | x |
| Santa_Cruz_Tenerife | | 28.47 | −16.25 | x |
| Tamanrasset_INM | | 22.79 | 5.53 | x |
| Teide | | 28.27 | −16.64 | |
| Eilat | Northeastern Africa | 29.50 | 34.92 | |
| El_Farafra | | 27.06 | 27.99 | |
| KAUST_Campus | | 22.30 | 39.10 | x |
| Medenine-IRA | | 33.50 | 10.64 | x |
| SEDE_BOKER | | 30.85 | 34.78 | x |
| Agoufou | Sahel | 15.35 | −1.48 | x |
| Banizoumbou | | 13.55 | 2.67 | x |
| Dakar | | 14.39 | −16.96 | x |
| DMN_Maine_Soroa | | 13.22 | 12.02 | x |
| IER_Cinzana | | 13.28 | −5.93 | x |
| Ilorin | | 8.48 | 4.67 | x |
| Ouagadougou | | 12.42 | −1.49 | x |
| Zinder_Airport | | 13.78 | 8.99 | x |
| Hamim | Arabian Peninsula | 22.97 | 54.30 | x |
| Kuwait_University | | 29.33 | 47.97 | x |
| Mezaira | | 23.10 | 53.75 | x |
| Mussafa | | 24.37 | 54.47 | x |
| Solar_Village | | 24.91 | 46.40 | x |
| Gandhi_College | Northern India | 25.87 | 84.13 | x |
| Jaipur | | 26.91 | 75.81 | x |
| Kanpur | | 26.51 | 80.23 | x |
| Karachi | | 24.95 | 67.14 | x |
| Lahore | | 31.48 | 74.26 | x |
| Dushanbe | Central Asia | 38.55 | 68.86 | x |
| IASBS | | 36.71 | 48.51 | |
| Cape_San_Juan | Caribbean | 18.38 | −65.62 | |
| La_Parguera | | 17.97 | −67.05 | |
| Trelew | Patagonia | −43.25 | −65.31 | |
| Tinga_Tingana | Australia | −28.98 | 139.99 | |

pared to Klose et al. (2021), who calculated a SW DRE of $-0.41\,\mathrm{W\,m^{-2}}$, with a DOD of 0.036 at 0.550 µm. Adjusting our SW DRE to match their DOD, through our DRE efficiency of $-12.75\,\mathrm{W\,m^{-2}}$, results in a similar value of $-0.45\,\mathrm{W\,m^{-2}}$. We have a more negative DRE efficiency compared to the value of $-11.55\,\mathrm{W\,m^{-2}}$ by Klose et al. (2021), despite our lower SSA (0.917 versus 0.953). This is probably attributable to modeling differences such as the

spectral dependence of DOD at UV–VIS wavelengths that we calculate as a band average or the assumed surface albedo.

On the other hand, our SW DRE at the TOA adjusted to the DOD of 0.03 (central value of the range from Ridley et al., 2016) is $-0.38\,\mathrm{W\,m^{-2}}$, which remains closer to the less cooling end of the range constrained by Kok et al. (2017) for $PM_{20}$ dust (approximately from $-0.80$ to $-0.15\,\mathrm{W\,m^{-2}}$, with a mean value near $-0.50\,\mathrm{W\,m^{-2}}$). This may partly be due to the warming effect of our higher fraction of absorb-

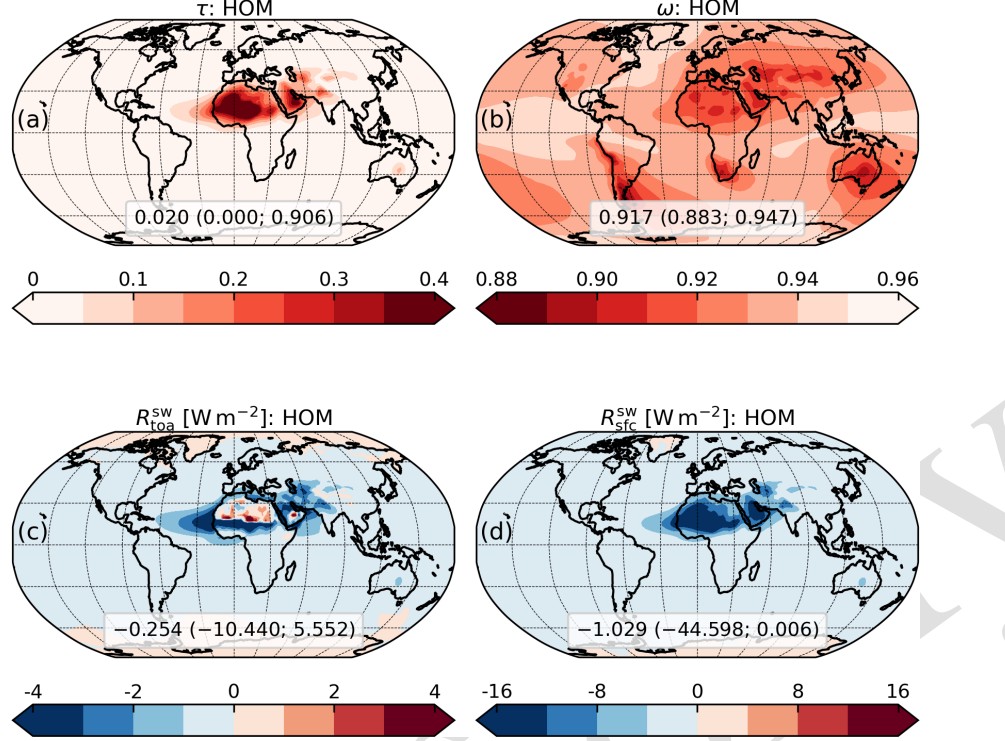

**Figure 6.** Annual mean optical depth ($\tau$; **a**), single scattering albedo ($\omega$; **b**) and direct radiative effect at the top of atmosphere ($R_{\mathrm{toa}}^{\mathrm{sw}}$; **c**) and at the surface ($R_{\mathrm{sfc}}^{\mathrm{sw}}$; **d**) from the control run with homogeneous composition (HOM). The optical properties are averaged over the spectral band of ModelE2.1 covering ultraviolet and visible wavelengths (0.30–0.77 μm), while the direct radiative effect is relative to the entire shortwave spectrum (0.30–4 μm). The extremes of the color bars are set to include the 1st and 99th percentiles of the mapped variables. Note that the color mapping for single scattering albedo (**b**) indicates lower values (i.e., stronger absorption) with more intense red. The global average along with the minimum and maximum (within parentheses) is also reported.

ing coarse particles (Fig. 7b), which are underrepresented by Kok et al. (2017) in the diameter range of 5–20 μm (according to Adebiyi and Kok, 2020) or are not included at all above 20 μm.

Our DRE efficiency is significantly more negative than the value of $-9.62\,\mathrm{W\,m^{-2}}$ estimated by Di Biagio et al. (2020), likely for the opposite reason: we have a lower fraction of absorbing coarse particles. Our SW DRE at the TOA of $-0.25\,\mathrm{W\,m^{-2}}$ equals their estimate ($-0.25 \pm 0.21\,\mathrm{W\,m^{-2}}$) despite their higher DOD of 0.026 at 0.550 μm. Di Biagio et al. (2020) extended the emitted PSD to diameters above 20 μm (up to 150 μm) by fitting airborne measurements from the FENNEC campaign (Ryder et al., 2013a, b). The warming effect of their giant particles, whose diameters are not included in our model, may partly explain their less negative DRE efficiency.

We highlight that the calculation of dust DRE at the TOA also depends upon the vertical distribution of dust, in addition to optical properties like DOD and SSA. Even under clear-sky conditions, the altitude of a dust layer modulates its absorption of solar radiation that is backscattered by the air molecules, especially over weakly reflecting surfaces, so that a higher dust layer results in a less negative SW DRE at the TOA even with a fixed column average DOD and SSA (Meloni et al., 2005). This introduces a poorly constrained uncertainty in the comparison among different model calculations, as evaluation of the dust vertical profile has typically received less attention in modeling studies.

### 3.1.2 Effect of externally mixed mineral components (EXT versus HOM)

In Fig. 8, we show the effect of the regionally varying soil mineral composition upon annual DOD and SSA in the UV–VIS band as well as upon the all-sky DRE at the TOA and at the SFC for the entire SW spectrum, assuming an external mixture of three radiatively active mineral components (EXT) with respect to HOM. Accounting for varying mineralogy enhances the extinction (Fig. 8a) while reducing the global absorption (Fig. 8b), with a global mean DOD contrast of $+3.6\,\%$ and the global mean SSA increasing from 0.917 to 0.936 (Table 3). In general, reducing absorption with a fixed mass would increase the SSA but leave the extinction almost unaffected (e.g., Fig. 9.7 of Mishchenko et al., 2002). In Fig. 9c, we show that the DOD increase due solely to distinguishing the CRI of the three mineral components would

**Table 3.** Global emission rate ($E$: annual total), global mass load ($L$: annual mean) and size-resolved lifetime ($T$) along with global-annual mean optical depth ($\tau$), single scattering albedo ($\omega$) and direct radiative effect at the top of atmosphere ($R_{toa}^{sw}$) and at the surface ($R_{sfc}^{sw}$) of model dust from the control run with homogeneous composition (HOM) and the mineral experiments with external (EXT) and internal (INT) mixing configurations for minerals. Estimates for diameters below 20 µm of the emission rate ($E_{20}$), mass load ($L_{20}$) and optical depth ($\tau_{20}$), calculated by assuming a constant sub-bin volume size distribution on the logarithmic scale (Appendix B2), are also reported. The optical properties are averaged over the spectral band of ModelE2.1 covering ultraviolet and visible wavelengths (0.30–0.77 µm), while the direct radiative effect is relative to the entire shortwave spectrum (0.30–4 µm). For completeness, the direct radiative effect relative to longwave bands is also reported ($R_{toa}^{lw}$ and $R_{sfc}^{lw}$ at the top of atmosphere and the surface, respectively). For each variable, the mean and standard error (within parentheses) over the simulation period (1991–2020) are reported.

| Variable | HOM | EXT | INT |
|---|---|---|---|
| $E$ (Tg yr$^{-1}$) | 6145.8 (47.9) | 6331.4 (48.9) | 6532.1 (63.2) |
| $E_{20}$ (Tg yr$^{-1}$) | 4030.5 (27.3) | 4152.0 (27.9) | 4283.6 (36.0) |
| $L$ (Tg) | 33.778 (0.316) | 35.031 (0.330) | 36.424 (0.409) |
| $L_{20}$ (Tg) | 31.296 (0.285) | 32.437 (0.300) | 33.740 (0.371) |
| $T$ (d) | Clay-1: 7.13 | Clay-1: 7.13 | Clay-1: 7.21 |
| | Clay-2: 7.13 | Clay-2: 7.13 | Clay-2: 7.21 |
| | Clay-3: 7.13 | Clay-3: 7.13 | Clay-3: 7.21 |
| | Clay-4: 7.13 | Clay-4: 7.13 | Clay-4: 7.21 |
| | Silt-1: 6.45 | Silt-1: 6.45 | Silt-1: 6.52 |
| | Silt-2: 4.89 | Silt-1: 4.91 | Silt-2: 4.96 |
| | Silt-3: 2.17 | Silt-1: 2.20 | Silt-3: 2.21 |
| | Silt-4: 0.43 | Silt-1: 0.43 | Silt-4: 0.44 |
| $\tau$ | 0.0199 (0.0002) | 0.0207 (0.0002) | 0.0215 (0.0003) |
| $\tau_{20}$ | 0.0197 (0.0002) | 0.0204 (0.0002) | 0.0213 (0.0002) |
| $\omega$ | 0.917 (0.001) | 0.936 (0.001) | 0.942 (0.001) |
| $R_{toa}^{sw}$ (W m$^{-2}$) | −0.254 (0.003) | −0.302 (0.003) | −0.337 (0.005) |
| $R_{toa}^{lw}$ (W m$^{-2}$) | 0.176 (0.002) | 0.183 (0.002) | 0.191 (0.002) |
| $R_{sfc}^{sw}$ (W m$^{-2}$) | −1.029 (0.010) | −0.975 (0.010) | −0.995 (0.013) |
| $R_{sfc}^{lw}$ (W m$^{-2}$) | 0.629 (0.006) | 0.646 (0.007) | 0.678 (0.008) |

be substantially negligible ($\lesssim 0.5\%$ of the control DOD in Fig. 6a over most source regions) compared to the DOD contrast in Fig. 8a. The latter, therefore, can be attributed to differences in aerosol mass between the EXT and HOM experiments. Despite identical calibration of the emitted mass as a function of wind speed, mass variations indicate a changed radiative feedback upon global emission and thus the mass load, whose contrast of +3.7 % is consistent with that of the global mean DOD. Also, in Fig. 9b we show the column mass load in the EXT experiment with respect to the smaller load in HOM (Fig. 9a): the regional distribution of the column load variations is well correlated with the DOD variations in Fig. 8a, which qualitatively confirms that the latter is nearly entirely due to the changed radiative feedback upon the dust mass. Under one possible feedback mechanism, the enhanced SW radiation reaching the surface (due to reduced absorption by the dust layer) increases boundary layer mixing and the vertical transport of strong winds aloft to the surface, thereby increasing dust emission (Miller et al., 2004; Pérez et al., 2006). This dependence is consistent with the greater SSA and mass load in EXT with respect to the HOM control run. However, a fuller investigation of the complicated dependence of surface wind speed and atmospheric

transport upon the dust DRE is beyond the scope of this paper.

The contrast of annual SSA due to varying mineralogy (Fig. 8b), according to the EXT coupling scheme compared to HOM, is remarkable in some locations. Among the high-DOD regions, in northern Africa (including plumes over the Mediterranean Sea), the Middle East and central Asia, we see positive SSA variations above 0.02 with peaks above 0.03 (e.g., in the northwestern Sahara), indicating a considerably higher scattering by dust in these regions. In contrast, we see smaller SSA variations (between 0.01 and 0.02) in the Arabian Peninsula and Australia along with almost negligible SSA variations (below 0.01) in the Sahel (and downwind over the Atlantic Ocean) and northern India. These regional variations reflect the distribution of the mineral components in the EXT experiment that carry different absorption properties. The host mixture, with a much more reflective (lower) IRI than our default dust (Table B1), is the most abundant of the three components over most dust regions (Fig. 10c). The static accretion, whose IRI is higher than our default IRI at most wavelengths, is particularly abundant in the Sahel (Fig. 10d), due to the strong emission of iron oxides in this area (Claquin et al., 1999; Gonçalves Ageitos et al., 2023),

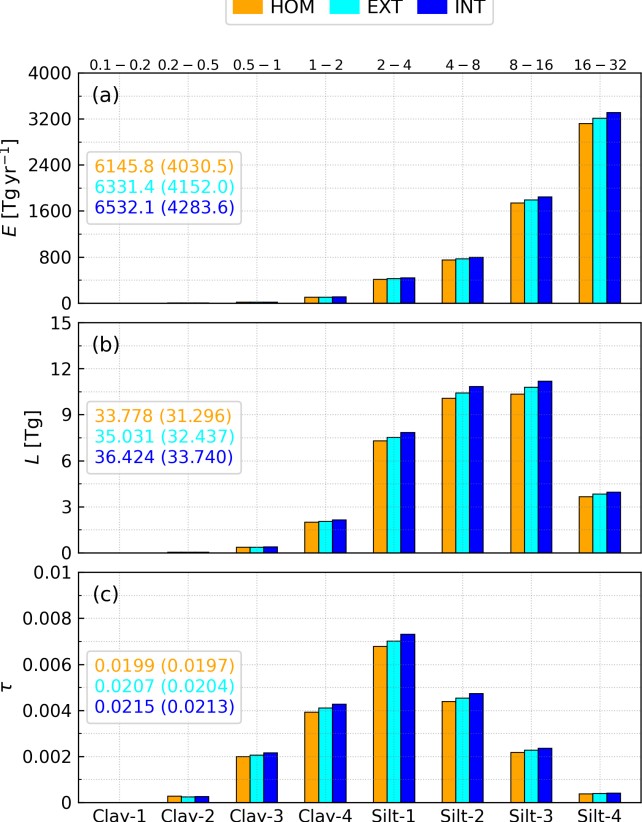

**Figure 7.** Size-resolved global emission rate ($E$: annual total; **a**), global mass load ($L$: annual mean; **b**) and global-annual mean optical depth ($\tau$; **c**) of model dust from the control run with homogeneous composition (HOM) and the mineral experiments with external (EXT) and internal (INT) mixing configurations. The optical depth is averaged over the spectral band of ModelE2.1 covering ultraviolet and visible wavelengths (0.30–0.77 μm). The ranges reported on the upper $x$ axis of the top panel indicate the diameters (μm) covered in each size bin (Table B2). Within each panel, the bulk values (sums over all bins) and our estimates for diameters below 20 μm (within parentheses) are reported for each experiment (Table 3).

where the SSA remains almost unchanged compared to the control experiment. This indicates that the effective absorption of a more abundant static accretion externally mixed with the background host amalgam resembles that of our default homogeneous dust. Free iron oxides have a comparatively negligible influence upon SSA due to their irrelevant optical depth (Fig. 10b: note the different color scale).

As shown in Fig. 8b, dust is more scattering in the EXT experiment compared to the control run (HOM). It is important to point out that, while the emphasized regional differences in SSA are a direct consequence of the varying mineral composition, the general increase in dust scattering depends upon the specific IRI that we use for unspeciated dust in the control experiment. The IRI retrieved by Sinyuk et al. (2003)

is just one of the possible IRI sets for dust available in the literature and in general is less absorbing than other widely used prescriptions (e.g., Patterson et al., 1977; Hess et al., 1998). According to the soil map of Claquin et al. (1999), along with our optical modeling based on DB19 and SZ15, dust aerosols are mainly composed by weakly absorbing host minerals in most regions, and the effect of absorbing iron oxides becomes relevant only for specific sources. The increase in dust scattering in the mineral experiment, therefore, can be interpreted as follows: currently available information about relative abundances and optical properties of minerals results in less absorbing dust particles compared to existing prescriptions for unspeciated dust, even those that have been traditionally considered highly reflective.

The negative contrast of the annual SW DRE at the TOA over most dust regions (Fig. 8c) is mainly determined by the increased SSA in the EXT experiment and is strengthened by the induced DOD increase. Over bright surfaces of prolific source regions (e.g., the Sahara and southern Arabian Peninsula), where the default SW DRE at the TOA is mostly positive (Fig. 6c), we see a strong warming reduction due to the higher SSA that causes lower absorption by the dust layer and thus more radiation reflected back into space. The negative SW DRE variations in these regions are quite uniform and large enough in magnitude ($\gtrsim 0.75\,\mathrm{W\,m^{-2}}$ in most areas) to modify the sign of the SW DRE by default dust in specific areas (compare Figs. 6c and S1a in the Supplement). Globally, the mineralogy effect of the three externally mixed components produces an additional cooling at the TOA of $-0.047\,\mathrm{W\,m^{-2}}$. We roughly estimate a relative contribution of 81 % from the SSA variations to this additional cooling by adjusting the SW DRE in the HOM control run to the DOD of the EXT experiment.

In Fig. 8d, the enhanced SSA tends to reduce the cooling at the SFC, because lower absorption leads to more forward-scattered radiation. This effect is opposed by the induced DOD increase, which actually causes greater cooling where the SSA contrast between the two experiments is comparatively small (e.g., in the Sahel). Globally, the externally mixed mineralogy effect produces a cooling reduction at the SFC of $0.054\,\mathrm{W\,m^{-2}}$.

### 3.1.3 Impact of the mixing configuration for minerals (INT versus EXT)

In Fig. 11, we evaluate the changes to DOD and SSA in the UV–VIS band, as well as to the all-sky DRE at the TOA and at the SFC for the entire SW spectrum, due to the assumed mixing configuration for minerals: INT (size-resolved internal mixing of all the minerals) with respect to EXT (external mixing of the three mineral components). The SSA is larger in the INT experiment (indicating more scattering), but the local contrast is small almost everywhere (below 0.0075; Fig. 11b) and the global mean contrast is only 0.006 (from 0.936 to 0.942; Table 3). This means that distributing iron

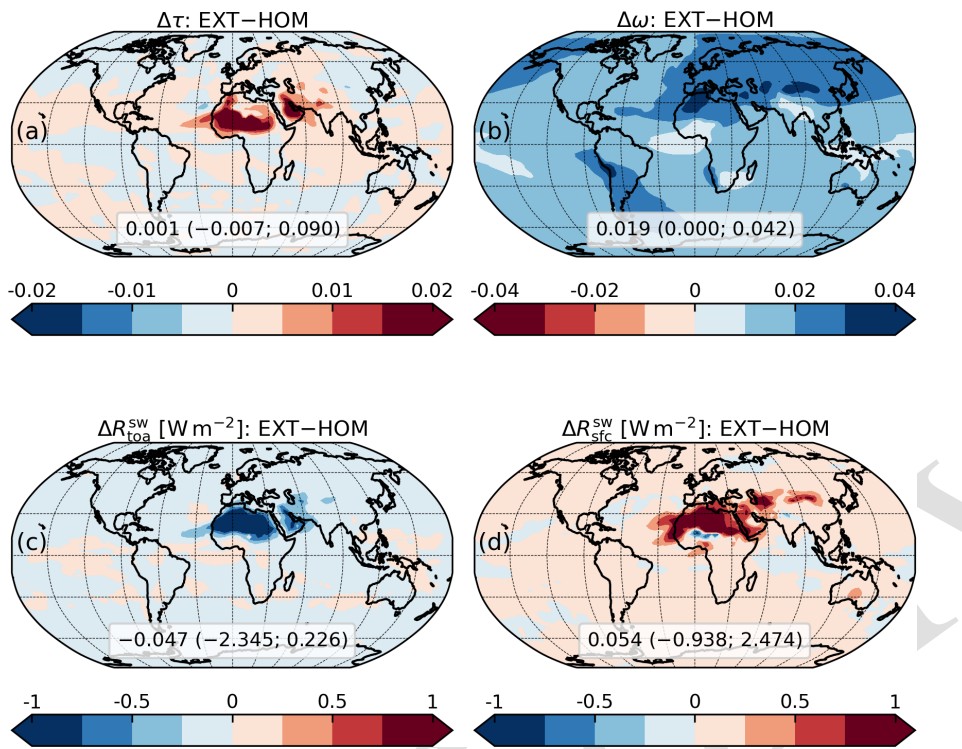

**Figure 8.** Same as Fig. 6 but showing the contrast between the mineral experiment with external mixing configuration (EXT) and the control run with homogeneous composition (HOM). Note that the color mapping for the single scattering albedo (**b**) indicates positive variations (i.e., increased scattering) with blue and vice versa.

oxides within a single mineral mixture, instead of incorporating them with a fixed proportion within only a fraction of the host minerals, slightly reduces dust absorption but does not substantially change the regional distribution of SSA. The
[5] SSA contrast between the two experiments is smallest downwind of source regions with high soil fractions of iron oxides, such as Sahel sources to the west of the Bodélé Depression as well as the Kalahari Desert, where the fractional abundance of accretions calculated by the model is largest (Fig. 15e of
[10] Perlwitz et al., 2015a). This suggests that our two mixing configurations result in similar absorption when the simulated fraction of iron oxides is large enough to substantially increase the IRI of the dynamic accretion (INT), according to Fig. 3, so that the latter resembles the external mixture of
[15] static accretion and the host amalgam in terms of absorptive power. (Note that the IRI of static accretion is always relative to 5 % of iron oxides by mass.)

The increased SSA determines a further widespread negative contrast of the SW DRE at the TOA over most dust
[20] regions (Fig. 11c), which is stronger where DOD increases are larger (Fig. 11a). The SW DRE variations are slightly lower in intensity compared to that generated by EXT versus HOM, with a ratio of global mean contrasts of 74 %, but they are still able to modify the distribution of the SW
[25] DRE at the TOA in specific areas (compare Fig. S1a and b). Globally, configuring the minerals as one dynamic internal

mixture (INT) generates an additional cooling at the TOA of $-0.035\,\mathrm{W\,m^{-2}}$ compared to the configuration with three externally mixed components (EXT). This is again mainly determined by the increased SSA ($\sim 64\,\%$), although the global
[30] mean DOD contrast ($+4.2\,\%$) makes a more relevant contribution than in EXT versus HOM.

The increase in global cooling at the TOA due to varying mineralogy (in both our EXT and INT experiments) partially contrasts with the results of SZ15. In that study, the reported
[35] dust SW DRE at the TOA is less negative in the experiments with explicit mineralogy ($-0.04$ and $-0.08\,\mathrm{W\,m^{-2}}$ in CAM4 and CAM5, respectively) compared to the corresponding experiments with unspeciated dust ($-0.14$ and $-0.33\,\mathrm{W\,m^{-2}}$ in CAM4 and CAM5, respectively). This decrease in global
[40] cooling at the TOA is associated with greater absorption indicated by a lower SSA in CAM5, but surprisingly with a higher SSA in CAM4. Using CAM5 with explicit mineralogy, Li et al. (2021) calculated a SW DRE at the TOA of smaller magnitude ($-0.18\,\mathrm{W\,m^{-2}}$) compared to our val-
[45] ues from the mineral experiments ($-0.30$ and $-0.34\,\mathrm{W\,m^{-2}}$ from EXT and INT, respectively; Table 3), despite their higher DOD of 0.03. The small magnitude of their DRE efficiency ($-5.94\,\mathrm{W\,m^{-2}}$) may be partly attributed to their quite low SSA of 0.89. (The latter, however, is inferred from the
[50] total aerosol SSA and is thus possibly affected by non-dust absorbing species.) Despite both these studies using the same

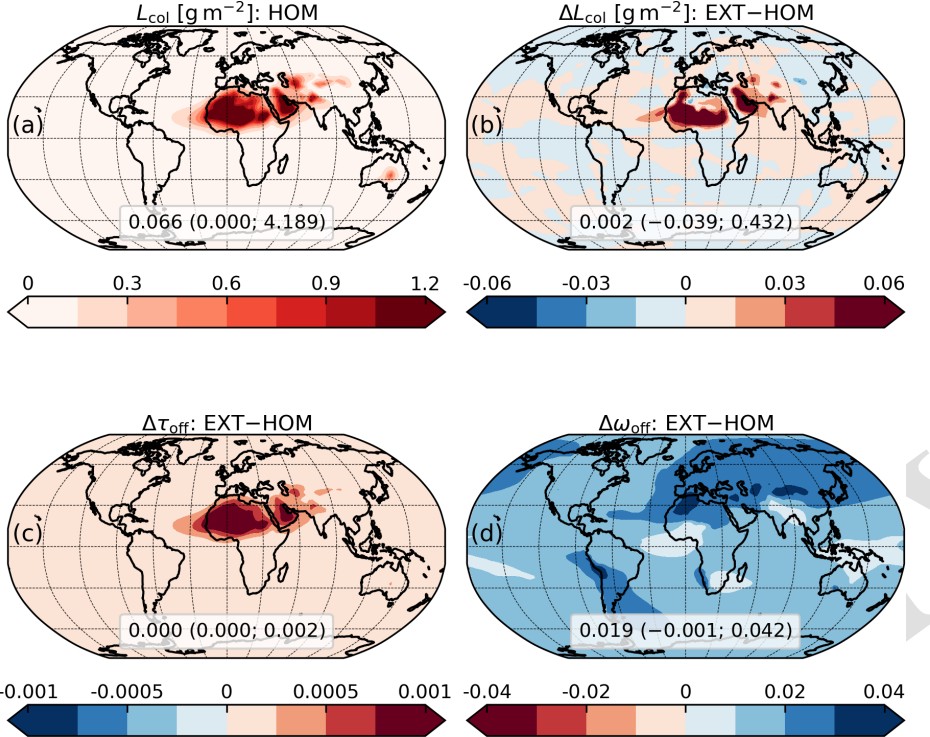

**Figure 9.** Top panels: annual mean column mass load ($L_{col}$) from the control run with homogeneous composition (HOM; **a**) and the contrast between the mineral experiment with external mixing configuration (EXT) and HOM **(b)**. Bottom panels: contrasts of annual mean dust optical depth ($\tau_{off}$; **c**) and single scattering albedo ($\omega_{off}$; **d**) of EXT with respect to HOM, calculated offline using the mass concentration of the three mineral components from HOM (in both cases) but distinguishing their refractive index in EXT. This offline calculation allows us to evaluate the effect of a changed absorption upon the optical properties without the contribution of a changed mass. The optical properties are averaged over the spectral band of ModelE2.1 covering ultraviolet and visible wavelengths (0.30–0.77 µm). The extremes of the color bars are set to include the 1st and 99th percentiles of the mapped variables. Note that the color mapping for single scattering albedo (**d**) indicates positive variations (i.e., increased scattering) with blue and vice versa. The global average along with the minimum and maximum (within parentheses) is also reported.

soil mineralogy map from Claquin et al. (1999) that we use in this work, differences in the regional and vertical distributions of the minerals and their optical properties may partly explain the contrasting results. For example, the absorption profile of dust, which is determined by the vertical distribution of different minerals and sizes, may affect the calculation of the dust SW DRE at the TOA, even with a fixed column-integrated DOD and SSA. This is because the relative height of absorbing and scattering layers modulates the amount of radiation absorbed in the column (Noh et al., 2016).

At the SFC (Fig. 11d), we see again the contrasting effect of SSA and DOD variations: while the SSA increase tends to reduce the cooling, the induced DOD increase softens this reduction and actually leads to a cooling increase where it is comparatively intense (e.g., plumes from the Sahel, over the Atlantic and the Mediterranean Sea). Globally, the INT scheme produces a cooling increase at the SFC of $-0.020\,\mathrm{W\,m^{-2}}$ with respect to EXT, which is primarily due to the enhanced DOD.

## 3.2 Comparison between model and AERONET dust optical properties

Figures 12, 13 and 14 show climatological monthly means of model dust SSA and DOD compared to the same variables retrieved for dusty scenes at the selected AERONET stations and months. Both model and AERONET optical properties are spectrally averaged over the UV–VIS band of ModelE2.1 (0.30–0.77 µm). We compare SSA to data from the AeroTAU4 set, while for DOD we use AeroTAU0: these data sets are defined based on the assumed minimum threshold for the AOD at 0.440 µm, as explained in Sect. 2.3. In Fig. 12, we quantify the dispersion of SSA across stations and months through the standard deviation of the monthly means. Similarly, in Figs. 13 and 14, we represent the variability of monthly SSA and DOD within each region and season through medians (solid bars) along with the 1st and 99th percentiles (error bars). For clarity, we refer to Northern Hemisphere seasons.

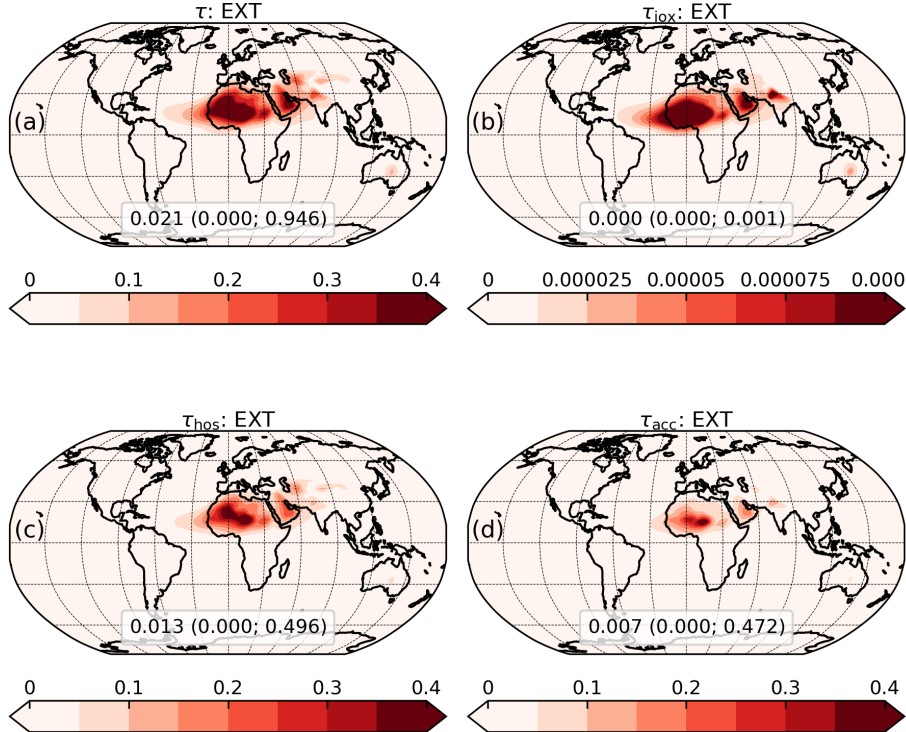

**Figure 10.** Annual mean dust optical depth ($\tau$) from the mineral experiment with external mixing configuration (EXT; **a**), decomposed into the contributions of free iron oxides ($\tau_{iox}$; **b**), the host amalgam ($\tau_{hos}$; **c**) and the static accretion ($\tau_{acc}$; **d**). The optical depth is averaged over the spectral band of ModelE2.1 covering ultraviolet and visible wavelengths (0.30–0.77 μm). The extremes of the color bars are set to include the 1st and 99th percentiles of the mapped variables. The global average along with the minimum and maximum (within parentheses) is also reported.

### 3.2.1 Spatiotemporal correlation of monthly single scattering albedo

The HOM model with globally homogeneous dust underestimates the AERONET SSA for most stations and months (Fig. 12a). This indicates that our IRI for unspeciated dust from Sinyuk et al. (2003) generally leads to excessive dust absorption compared to AERONET (see Sect. 4.3), except for some stations within the Sahel, the region better represented by the IRI from Sinyuk et al. (2003), where model SSA values around 0.92 closely align with AERONET values. Other exceptions to the low bias of model SSA, for some spring and summer months in northern India, may be partly due to contamination of our dusty AERONET scenes by absorbing carbonaceous species (Go et al., 2022). In addition, our homogeneous dust experiment produces a range of SSA across stations and months that is lower than the AERONET range: the modeled and retrieved standard deviations are $\sigma = 0.006$ and $\sigma = 0.011$, respectively. Except for some SSA values above $\sim 0.92$ at northwestern African stations, such as Capo_Verde and the sites in the Canary Islands (Figs. 5 and 6b), the model SSA ranges approximately from 0.90 to 0.92, while the AERONET SSA ranges from 0.90 to 0.95. In the HOM experiment, the SSA variability is solely attributable to variations in the size distribution. A model PSD underrepresenting the real size variability, for example, may contribute to the mismatch between modeled and retrieved SSA ranges. However, our analysis suggests that part of the AERONET SSA range missed by the HOM model can be explained by missing variations in mineralogy.

Figure 12b and c TS3 show the model SSA in the EXT and INT experiments, respectively, compared to the AERONET SSA. The first evident result is that accounting for varying composition increases the model SSA values (which corresponds to greater scattering) and enlarges the SSA range, improving overall the agreement with AERONET. The EXT scheme produces SSA values roughly between 0.91 and 0.95 with the same SSA dispersion of the INT scheme ($\sigma = 0.013$), the latter reaching slightly higher SSA (up to 0.96) except for some spring and winter months in the Sahel (consistent with Fig. 11b). This is more clearly visible in Fig. 12d, where we directly compare the EXT and INT experiments: only in 1 winter month is the SSA slightly lower in INT compared to EXT. This corroborates our argument from Sect. 3.1.3: if the fraction of iron oxides becomes large enough, the INT configuration with dynamic accretion results in similar (or higher) absorption compared to the EXT configuration, where the static accretion is externally mixed with the host mixture.

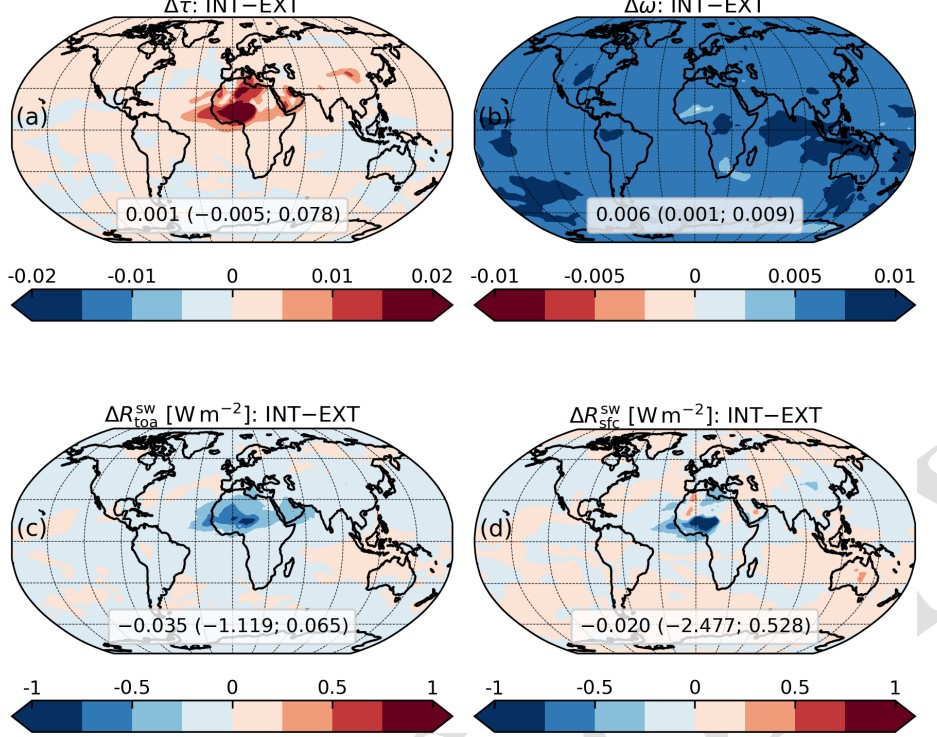

**Figure 11.** Same as Fig. 6 but showing the contrast between the mineral experiments with internal (INT) and external (EXT) mixing configurations. Note that the color mapping for single scattering albedo (**b**) indicates positive variations (i.e., increased scattering) with blue and vice versa.

This SSA comparison demonstrates that prescribing a globally uniform IRI for dust, as most climate models still do, makes it challenging to reproduce the spatiotemporal variability of the observed dust SSA and inevitably generates biases across different regions. Consideration of mineral composition, regardless of our chosen mixing configuration, leads to better reproduction of the range of AERONET dust absorption across different locations and months, which suggests that the mismatch between the SSA ranges from the HOM model and AERONET is partly due to the effect of varying mineral composition. Nonetheless, the model–AERONET correlation improves only very slightly in the mineral experiments (from $r = 0.24$ in HOM to $r = 0.29$ and $r = 0.27$ in EXT and INT, respectively): this increase calculated for $N = 91$ values is not statistically significant according to a two-sided 95 % confidence interval. The moderate correlations in the mineral experiments suggest remaining model limitations in correctly simulating the mineral composition of dust aerosol and its absorption properties (Sect. 4).

### 3.2.2 Regional and seasonal comparison of monthly single scattering albedo and optical depth

The regional and seasonal SSA comparison, in Fig. 13, corroborates our interpretation of Fig. 12. In almost all the regions and seasons, the AERONET SSA is higher than the SSA of the HOM model and shows larger variability across different regions, with individual station values from ∼ 0.90 in northern India during spring to peaks exceeding 0.95 in the Arabian Peninsula during summer. In contrast, the HOM model SSA ranges approximately from 0.90 to 0.93 at the AERONET sites, consistent with Figs. 6b and 12a. Also, the inter- and intra-seasonal SSA variability within each region is generally higher in AERONET than in our HOM experiment in most regions. For example, note the remarkable drop in retrieved SSA from fall to winter in the Sahel that is absent in the HOM experiment or the larger AERONET error bars in northeastern Africa, the Sahel during summer and winter, the Arabian Peninsula during summer and northern India.

The evaluation is improved by representing spatial and temporal variations of dust composition, which first of all generates a higher model SSA, with INT values consistently higher than EXT values (as already seen in Figs. 11b and 12d). Note that, in the Sahel during winter, both the EXT and INT experiments reproduce the wintertime increase in absorption indicated by AERONET, in contrast to the HOM experiment. The varying mineralogy effect also generates larger regional and seasonal SSA variability and thus generally improves the agreement with AERONET. However, evident discrepancies between the mineral-speciated models and the AERONET SSA remain: one example is the mineral experiments overestimating the AERONET SSA in north-

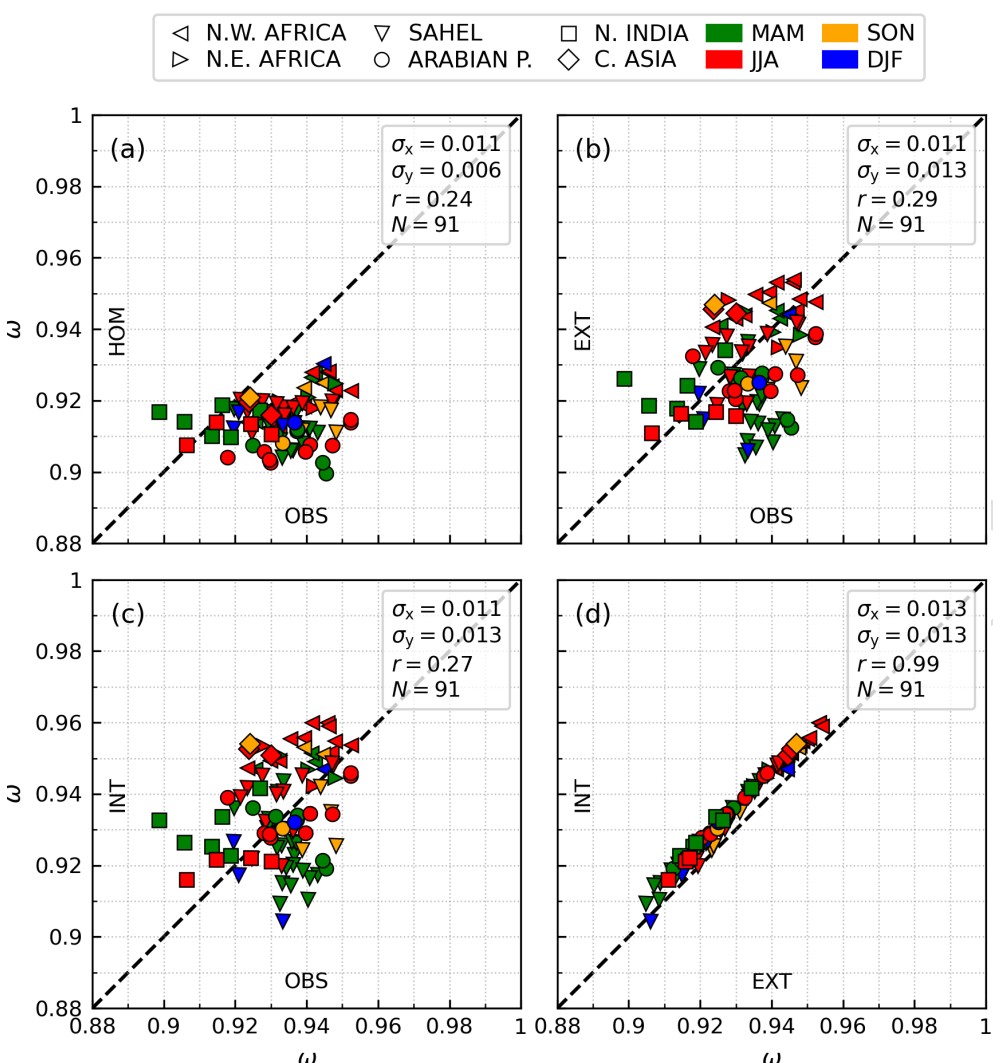

**Figure 12.** Climatological monthly mean single scattering albedo ($\omega$) from model experiments compared to AERONET monthly retrievals at the selected stations and months from the AeroTAU4 data set, labeled by region (Fig. 5) and season (Northern Hemisphere spring: MAM; summer: JJA; fall: SON; winter: DJF). The single scattering albedo is averaged over the spectral band of ModelE2.1 covering ultraviolet and visible wavelengths (0.30–0.77 μm). AERONET monthly means (OBS) are separately compared to the control run with homogeneous composition (HOM; **a**) and the mineral experiments with external (EXT; **b**) and internal (INT; **c**) mixing configurations; the two mineral experiments are also directly compared (**d**). The standard deviation of the monthly means ($\sigma$), Pearson correlation coefficient ($r$) and number of monthly values ($N$) are also reported.

ern Africa and underestimating it in the Sahel in most seasons (see Sect. 4.1). Evaluation of the annual cycle does not present a clear conclusion: the model simulates slightly more scattering dust during summer in northern Africa and the Sahel, but this tendency is less clear in the AERONET observations, whose annual cycle is different for each region.

AERONET DOD is underestimated by the model in most regions and seasons (Fig. 14), which is consistent with our global DOD underestimating the observational constraint provided by Ridley et al. (2016), although we find good agreement in the Sahel and Arabian Peninsula, possibly due to a better modeled emission in these source regions. De-

spite the partial mismatch of values, we highlight the good agreement between modeled and observed annual cycles: the observed DOD peaking during summer or spring, before going down in fall and winter, is generally well reproduced by the model. Model DOD differences resulting from different coupling schemes are not as important as for SSA. We see small DOD contrasts due to a differing radiative feedback upon dust emission and mass load (consistent with Figs. 8a and 11a).

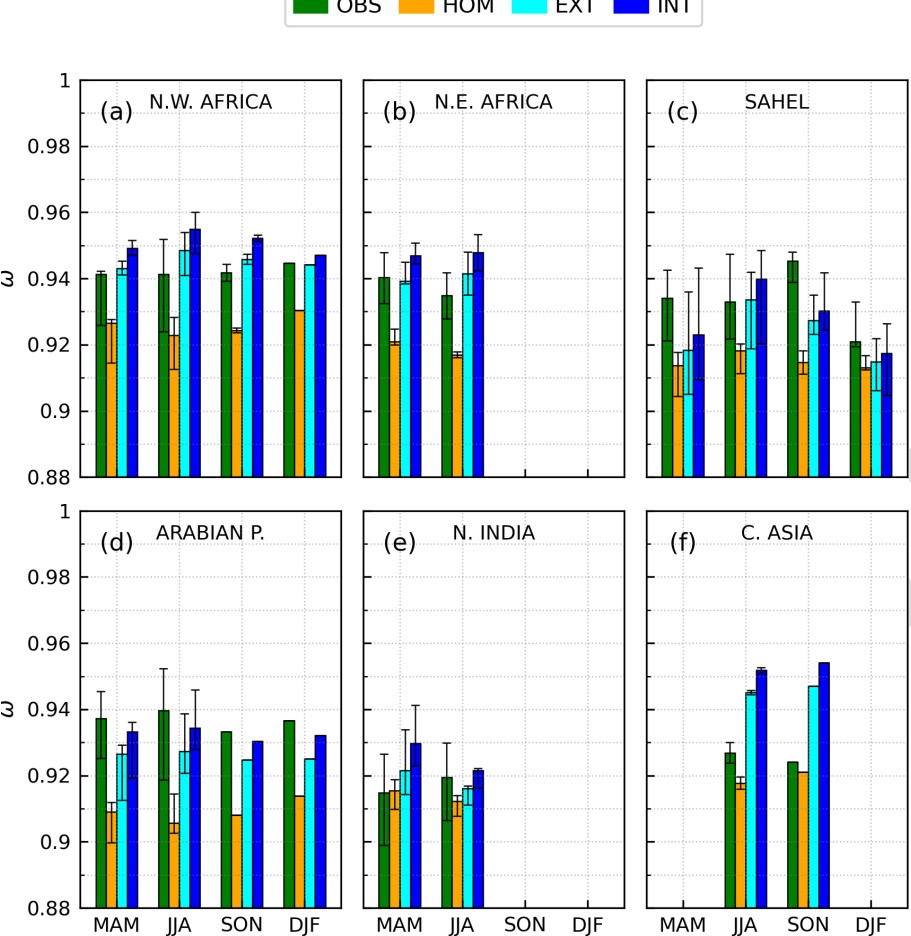

**Figure 13.** Climatological monthly mean single scattering albedo ($\omega$) from the control run with homogeneous composition (HOM) and the mineral experiments with external (EXT) and internal (INT) mixing configurations compared to AERONET monthly retrievals (OBS) at the selected stations and months from the AeroTAU4 data set, grouped per region (Fig. 5) and season (Northern Hemisphere spring: MAM; summer: JJA; fall: SON; winter: DJF). The single scattering albedo is averaged over the spectral band of ModelE2.1 covering ultraviolet and visible wavelengths (0.30–0.77 µm). The solid bars represent medians along with the 1st and 99th percentiles (error bars) of the monthly means within each region and season.

## 4 Sources of uncertainty

### 4.1 Inaccuracy of the soil mineralogy map and model processes

Soil fractions of the eight minerals simulated here are characterized by only a few hundred measurements worldwide. Claquin et al. (1999) circumvented this scarcity by characterizing the soil mineral composition using comparatively abundant surveys of soil types that are available globally, assuming that each type has a characteristic mineral content: the "mean mineralogy" assumption. The inferred mineral fractions are uncertain both because of the limited measurements available to establish the composition of each soil type and the neglect of mineral variations within a single soil type across different regions. Journet et al. (2014) revisited the soil mineralogy, increasing the number of measure-

ments and using the soil unit descriptor to extend the mineral fractions to a global scale. Simulations using both representations of soil composition perform comparably, although the evaluation is limited by the sparsity of atmospheric measurements of aerosol mineralogy (Gonçalves Ageitos et al., 2023). The soil fractions of some radiatively important minerals like iron oxides are additionally uncertain due to qualitative metrics that describe their abundance, like the "redness" of the soil. All these challenges are currently being addressed through spaceborne hyperspectral imaging from the International Space Station as part of NASA's Earth Surface Mineral Dust Source Investigation (EMIT) mission that will provide billions of mineral identifications based on spectroscopic measurements across VIS and NIR wavelengths (Green et al., 2020). This project will be especially effective

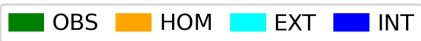

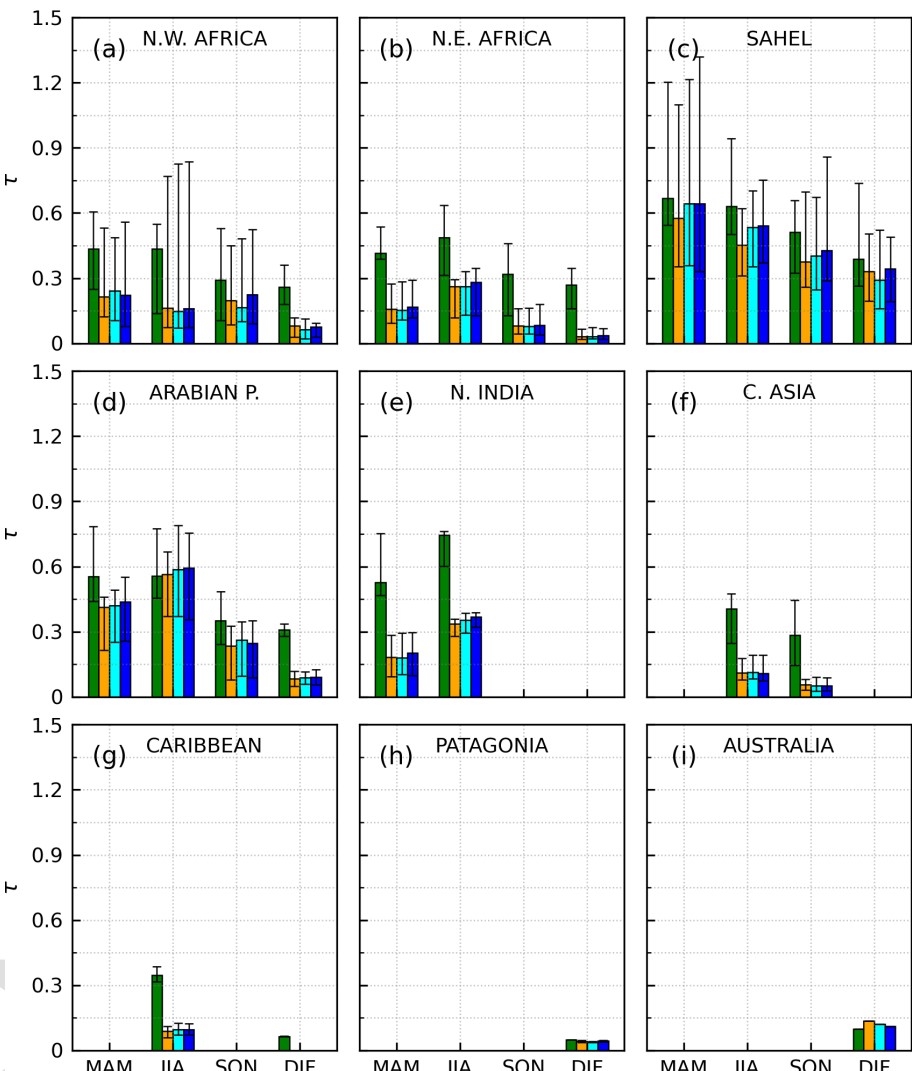

**Figure 14.** Same as Fig. 13 but showing the comparison between model and AERONET dust optical depth ($\tau$) at the selected stations and months from the AeroTAU0 data set.

at detecting iron oxides like hematite and goethite that have prominent absorption features in the UV and VIS bands.

Other model errors of mineralogy arise from the limited spatial resolution of the ESM. This is a special problem for emission, which depends upon the relation of dust sources to the local surface winds that are partly steered by topography, which may be smoothed in the model. This leads to calculations of emission by different ESMs with varying regional emphases and biases (Kok et al., 2021). The model resolution also degrades the spatial detail of the soil unit and soil type atlases, blurring the mineral fractions. Additional modeling challenges are the emitted PSD that we assume to be globally uniform (except for slight variations across the silt bins

due to geographically varying mineral composition; Perlwitz et al., 2015a), possible transport errors that may cause plumes from different source regions to be mixed with biased proportions and the possibly excessive removal of the coarsest particles during transport resulting in underestimated coarse fractions far from sources (e.g., van der Does et al., 2016).

The aerosol mineral composition calculated by the model is influenced by all these uncertainties, which may lead to biased relative abundances of minerals and thus eventually impact the absorption properties of the dust mixture at specific grid boxes. For example, during spring, our mineral experiments calculate much lower SSA in the Sahel compared to northern Africa, whereas AERONET reports simi-

lar SSA (Fig. 13). This contributes to the slightly excessive ranges and moderate correlations of model SSA compared to AERONET (Fig. 12b and c). Our model contrast results from higher emitted fractions of iron oxides in the Sahel than in the Sahara (Fig. 12 of Perlwitz et al., 2015a), which is consistent with the composition measurements by DB19 (except in the Bodélé Depression). Given that AERONET photometers scan the entire column, part of the error could be inaccurate transport of model dust between these two regions in specific seasons that results in insufficient mixing of dust plumes carrying different absorption properties.

## 4.2 Uncertainty in the optical modeling of minerals

One source of uncertainty in our optical modeling of minerals is the definition of a single globally uniform host mixture, whose CRI is calculated by composing the CRIs of host minerals and then averaging over the dust aerosol samples of DB19. Our neglect of spatial and temporal variations in the composition of the host amalgam is a helpful modeling simplification but may bias the regional distribution of model SSA. For example, the model emission at clay sizes is dominated by the phyllosilicate illite within the Sahara, while higher fractions of clay-sized kaolinite are emitted in the Sahel (Fig. 10 of Perlwitz et al., 2015a). Because illite is more absorbing than kaolinite (Fig. 2b), this variation of emitted fractions of the two minerals would result in a (small) regional contrast of absorption by the host mixture that is neglected in our model. Moreover, IRI differences among host minerals (e.g., smectite and illite have a larger IRI than quartz and calcite at UV–VIS wavelengths; Fig. 2b) challenge our use of the VM mixing rule for the host mixture, whose validity as a limit of the Lorentz–Lorenz mixing rule depends upon vanishing CRI differences among the constituent minerals.

Uncertainty in the IRI retrievals from DB19, as well as in our polynomial representation of their empirical relations to the iron oxide content (Fig. 3), directly impacts our calculation of the IRI for accretions. DB19 retrieved an "effective" IRI in each dust aerosol sample (using the measured volume size distribution) that is representative of homogeneous spherical particles within a bulk size range up to $\sim 10\,\mu m$ in diameter. One uncertainty is that we use the same relationship between IRI and iron oxides for all our size bins, neglecting any possible size effects upon the inversion calculation. Our assumption is partially supported by DB19, as they reported a negligible dependence of the retrieved IRI upon the settling of the coarser particles in each sample during the experiment duration (2 h). This suggests that the resulting reduction in the size range did not significantly affect the relation of the retrieved IRI to the fractional amount of iron oxides (measured by accumulating the filtered aerosol mass for the entire experiment duration). However, a systematic validation would be useful for strengthening the use of the same empirical relationship for different size ranges, also including diameters above $10\,\mu m$.

## 4.3 Uncertainty in AERONET single scattering albedo and size retrievals

AERONET retrievals are subject to uncertainties that affect the model evaluation in Figs. 12 and 13. The error of the individual hourly SSA retrievals is $\sim 0.03$ (Level 2.0), due to measurement errors and uncertain inputs to the inversion algorithm (Dubovik et al., 2002; Sinyuk et al., 2020). Consequently, the resulting monthly mean SSA that averages over multiple hourly retrievals is uncertain by $\lesssim 0.004$. Further uncertainty in the AERONET monthly SSA may arise from an inaccurate selection of dusty scenes. Despite our strict conditions for isolating dust events, even a small contamination by residual carbonaceous absorption (i.e., by BC and BrC) could still affect the observed SSA at specific stations and months (Schuster et al., 2016). This error is difficult to characterize.

Finally, in addition to an excessively absorbing IRI from Sinyuk et al. (2003) (Sect. 3.2.1), our coarse size distribution may contribute to the general low bias of the HOM model SSA compared to AERONET (Fig. 12a), as larger particles have in general a lower SSA. Despite our fine and coarse fractions of mass load being consistent with the semi-observational constraint by Adebiyi and Kok (2020) as discussed in Sect. 3.1.1, our model systematically represents higher fractions of coarse dust than indicated by AERONET in the selected dusty scenes (for diameters above $8\,\mu m$ according to Fig. S4). However, while the AERONET SSA retrievals are heavily constrained by sky radiance measurements from the almucantar scan (e.g., Schuster et al., 2016), the AERONET inversion algorithm assumes that the volume size distribution falls to zero near diameters of $30\,\mu m$ (Dubovik et al., 2002). In principle, in addition to missing larger particles (e.g., Ryder et al., 2019), the assumed cutoff could also lead to underrepresentation of smaller particles with diameters approaching $30\,\mu m$ if such coarse particles remain suspended long enough to be detected by the instrument. To date, this potential uncertainty in the AERONET size retrieval has not been quantified. Therefore, we cannot assess to what extent our underestimation of AERONET SSA in the HOM case results from excessive coarse particles in the model.

## 5 Conclusions

In this work, we couple the spatially and temporally varying aerosol mineral composition, derived from regionally varying soil mineralogy (Claquin et al., 1999), to SW radiation calculations in the NASA GISS ModelE2.1. Measurements of dust particles that might guide the model representation of mineral mixtures are available only from limited field campaigns (e.g., Kandler et al., 2009; Panta et al., 2023). De-

riving the particle CRI from the indices of the constituent minerals requires the application of approximate theoretical mixing rules, whose validity is difficult to assess. Moreover, IRI estimates for iron oxides, which dominate dust absorption at solar wavelengths, are highly uncertain (Zhang et al., 2015). As an alternative approach, we link the dust absorption to mineral composition according to empirical relationships proposed by DB19 that relate the dust IRI at solar wavelengths to the mass fraction of iron oxides. We assess the effect of varying composition upon dust optical properties (DOD and SSA) in the UV–VIS band (0.30–0.77 µm) and the DRE summed across the SW spectrum. To evaluate the sensitivity of our results to the mixing state of minerals, we use two alternative mineral–radiation coupling schemes. For the INT experiment, all the minerals are blended into a single size-varying internal mixture whose IRI is proportional to the spatially and temporally varying iron oxide mass fraction. In contrast, for the EXT experiment, we define an external mixture of three radiatively active components. The first is a homogeneous amalgam of the non-iron oxide minerals, based on aerosol volume fractions from DB19 combined with the CRIs of these minerals from SZ15. Our second component internally mixes this amalgam with iron oxides. (A third component containing pure iron oxides makes a negligible contribution to the total DOD.) We evaluate the model SSA and DOD against observations from AERONET filtered for dust events. We define a new technique for selecting scenes dominated by dust in AERONET, based on filtering conditions applied to retrievals of size (FVF) and absorption properties (SSA and IRI) of both the dust and non-dust aerosol species at solar wavelengths (see also Gonçalves Ageitos et al., 2023).

All versions of our dust model underestimate global dust extinction while overestimating the dust mass inferred by Kok et al. (2017), who used a combination of satellite, in situ and modeling constraints. Our models exhibit a fine fraction of dust mass that is consistent with Kok et al. (2017), while we have a greater coarse fraction. This indicates that the extinction efficiency of our particles is too small. We attribute this bias to our assumption of spherical particle shape. Kok et al. (2017) calculated that approximating the highly irregular shapes of natural dust particles (Kalashnikova and Sokolik, 2004; Huang et al., 2020) with triaxial ellipsoids (Meng et al., 2010) leads to a significantly larger extinction efficiency, mostly due to the higher cross-sectional area of ellipsoids with respect to volume-equivalent spheres (Huang et al., 2023). On the other hand, our use of Mie theory to compute optical properties (Appendix B) is consistent with the assumption of spherical shape in the inversion calculation of dust IRI by DB19. While we address the shape effect upon the optical properties and DRE of unspeciated dust in an upcoming study, we would need to assess its impact upon the empirical relations of the dust IRI to the fractional mass of iron oxides before extending our approach to non-spherical particles.

In the HOM control experiment that assumes globally homogeneous composition, we use a dust IRI retrieved at locations that are downwind of sources from both the Sahel and the Sahara (Sinyuk et al., 2003). This IRI is often taken by modelers as globally representative (e.g., Miller et al., 2006; Yoshioka et al., 2007). Compared to the control case, the mineral experiments generally exhibit higher scattering, with the global SSA increasing from 0.917 in HOM to 0.936 and 0.942 in EXT and INT, respectively, as well as a slightly higher DOD due to the changed radiative feedback upon dust emission (from 0.020 in HOM to 0.021 and 0.022 in EXT and INT, respectively). The mineral-induced SSA contrasts emphasize some regional differences (e.g., between the Sahara and the Sahel) that are mainly determined by the varying abundance of iron oxides. Simulated iron oxide fractions result in similar absorption compared to our unspeciated dust in only a few regions like the Sahel. The regional variations in SSA and DOD perturb the spatial distribution of the DRE, both at the TOA and at the SFC, with a potential impact upon the regional climate. Globally, the increased SSA is the main contributor to greater cooling at the TOA: the SW DRE varies from $-0.25$ in HOM to $-0.30$ and $-0.34 \, \mathrm{W\,m^{-2}}$ in EXT and INT, respectively.

Dust optical properties and the DRE are sensitive to the mixing state of minerals, with a consistent increase in both SSA and DOD, along with a larger global cooling at the TOA, when all minerals (including iron oxides) are amalgamated into a single internal mixture (INT) compared to accreting iron oxides within only a fraction of the host minerals (EXT). Compared to the HOM control run, the change to the SW DRE at the TOA in INT ($-0.083 \, \mathrm{W\,m^{-2}}$) is nearly 2 times larger in magnitude than in EXT ($-0.047 \, \mathrm{W\,m^{-2}}$). The SW effect is offset by the positive LW DRE (Table 3; Fig. S2a and b for EXT and INT, respectively), so that the net DRE at the TOA has smaller magnitudes: $-0.12$ and $-0.15 \, \mathrm{W\,m^{-2}}$ for the EXT (Fig. S3a) and INT (Fig. S3b) experiments, respectively, compared to $-0.08 \, \mathrm{W\,m^{-2}}$ for the HOM control run. The varying mineralogy also leads to a weaker cooling at the SFC with a SW DRE from $-1.03$ in HOM to $-0.97$ and $-0.99 \, \mathrm{W\,m^{-2}}$ in EXT and INT, respectively. At the SFC, INT results in stronger cooling than EXT due to the effect of the increased DOD.

The explicit accounting for aerosol mineral composition and its spatiotemporal variations increases the model SSA, reducing its bias in comparison to AERONET observations, and also enlarges the range of SSA variations across different stations and months, further improving the agreement with AERONET. In contrast, assuming homogeneous composition according to the specific IRI from Sinyuk et al. (2003) results in excessive absorption with respect to AERONET in most stations and months without reproducing the full variability of retrieved SSA. One potential uncertainty is that Sinyuk et al. (2003) retrieved dust IRI at only two UV wavelengths and inferred values from 0.331 to 0.670 µm by log-linearly interpolating with AERONET IRI retrievals. We fur-

ther extend these IRI values to encompass the entire UV–VIS band in our model. This extrapolation may not accurately capture IRI variations across VIS wavelengths, potentially overemphasizing absorption.

The reduction in the model absorption bias with respect to AERONET in the mineral experiments is a consequence of two factors. First, dust aerosol absorption largely results from the varying fraction of iron oxides, given the comparatively weak absorption by the other simulated minerals at solar wavelengths. Second, hematite is often observed together with goethite (Formenti et al., 2008), another iron oxide that is less absorbing than hematite (Bedidi and Cervelle, 1993). We take into account the radiative effect of goethite by using the DB19 empirical relationships between dust IRI and the combined mass fraction of both iron oxides, thus interpreting our soil hematite fraction prescribed by Claquin et al. (1999) more generally as a less absorbing iron oxide mixture that includes goethite. Our reduced absorption bias suggests that a realistic representation of the dust absorption at UV–VIS wavelengths must be founded upon accurate prescription of the separate hematite and goethite fractions in soils along with the modeling of their individual radiative effects. These improvements may also lead to a stronger correlation between model and AERONET SSA at specific locations and months while preserving the agreement of the spatial and temporal ranges.

Measurements to evaluate the modeled aerosol mineral fractions are limited (Perlwitz et al., 2015b; Pérez García-Pando et al., 2016), which results in uncertain estimates of measurement climatologies. In contrast, modelers are on the verge of greatly improved knowledge of soil mineral content from the NASA EMIT hyperspectrometer that is currently measuring VIS and NIR surface reflectances from the International Space Station. The few hundred soil analyses that are the basis for current soil atlases like Claquin et al. (1999) and Journet et al. (2014) will be supplemented with over a billion retrievals of soil minerals, providing unprecedented knowledge of source composition with both greater regional detail and global extent (Green et al., 2020).

Despite the remaining discrepancies with respect to AERONET observations, this work represents a promising step towards a refined modeling system capable of incorporating the huge amount of information about soil composition that will soon be provided by EMIT. This could allow the NASA GISS ModelE2.1 to achieve a more realistic representation of dust as a composite of different minerals that would enable an improved assessment of the dust DRE and its impacts upon regional and global climate.

## Appendix A: Derivation of the imaginary index of free iron oxides

In the EXT scheme, we need to assign a CRI to the radiatively active component consisting solely of crystalline

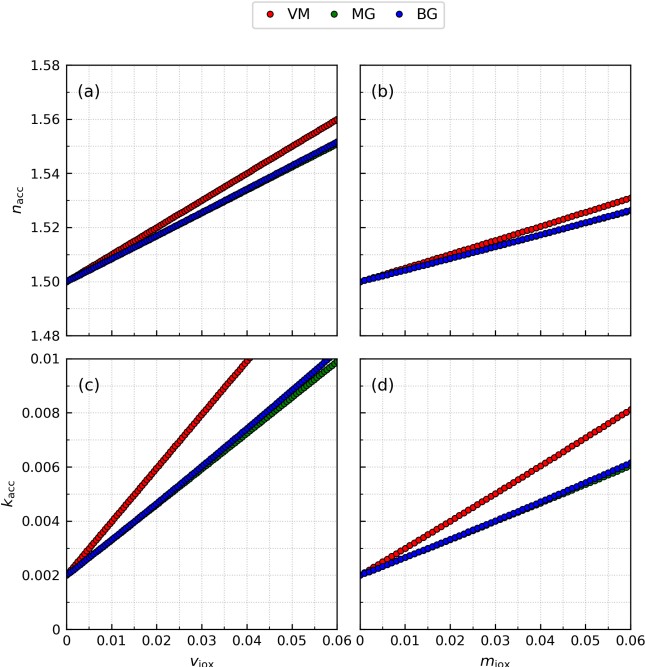

**Figure A1.** Test calculation of the dependence of real ($n_{acc}$; **a** and **b**) and imaginary ($k_{acc}$; **c** and **d**) indices of an accretion-like composite particle upon the volume ($v_{iox}$; **a** and **c**) and mass ($m_{iox}$; **b** and **d**) fractions of an iron oxide-like inclusion embedded in a homogeneous host-like medium according to different mixing rules (volume-weighted mean: VM; Maxwell Garnett: MG; Bruggeman: BG) in the limit of small fractions of the inclusion (below 0.06). The real indices of the inclusion and host are, respectively, set to $n_{iox} = 2.5$ and $n_{hos} = 1.5$, the imaginary indices to $k_{iox} = 0.2$ and $k_{hos} = 0.002$ and the mass densities to $\rho_{iox} = 5\,\mathrm{g\,cm}^{-3}$ and $\rho_{hos} = 2.5\,\mathrm{g\,cm}^{-3}$.

or free iron oxides. Here, we describe the inversion procedure for calculating the IRI of iron oxides consistent with the empirical relationships of DB19 that we compare to literature values corresponding to iron oxide minerals like hematite and goethite (Fig. 4). As a first step, we must assume a mixing rule to link the IRI of dust particles to that of its constituent minerals. Figure A1 shows that commonly used mixing rules such as VM (volume-weighted mean), MG (Maxwell Garnett) and BG (Bruggeman) are approximately linear for small fractions of iron oxides (as measured by DB19). (VM is exactly linear for any fraction of iron oxides.) This linearity suggests that we can derive the IRI of iron oxides by matching an analytic expression of a mixing rule (that we derive below) to the linear regression of the empirical relationships from DB19 between the particle IRI and the iron oxide fraction.

In general, the MG mixing rule (Markel, 2016) gives the effective complex permittivity $\epsilon = (n + ik)^2$, where $n$ and $k$ are the RRI and IRI, respectively, of a composite (in our case accretion) starting from the permittivities of a homogeneous host medium and small inclusions (iron oxides). In the linear

limit for small volume fractions of iron oxides ($v_{iox}$), the MG expression is identical to BG (Markel, 2016) and is given by CE1

$$\begin{cases} \epsilon_{acc} = \epsilon_{hos} + 3\epsilon_{hos} X v_{iox} \\ X = \frac{\epsilon_{iox} - \epsilon_{hos}}{\epsilon_{iox} + 2\epsilon_{hos}} \boxed{TS4} \end{cases}, \tag{A1}$$

whose imaginary component is CE2

$$2n_{acc}k_{acc} = 2n_{hos}k_{hos} + 3I v_{iox}, \tag{A2}$$

where the factor $I$ is the imaginary part of the product $\epsilon_{hos} X$. From Eq. (A2), it is evident that MG couples the RRI and IRI of the accretion, which in turn depend upon coupled RRIs and IRIs of both the host medium and iron oxides according to a nonlinear relation. Therefore, to derive an equation only for the IRIs, an independent constraint upon the RRIs is required. For simplicity, we use Eq. (2) (i.e., the VM rule) as the linear relation among the RRIs of accretion, host medium and iron oxides: as already discussed in Sect. 2.2.3, for small fractions of iron oxides, different mixing rules result in negligible differences in the accretion RRI. Substituting Eq. (2) into Eq. (A2) leads to

$$k_{acc} = \frac{n_{hos}k_0 + (3/2)I v_{iox}}{n_{hos} + (n_{iox} - n_{hos}) v_{iox}}, \tag{A3}$$

where we indicate the host medium IRI with $k_0$ to highlight that here it is in general different from the IRI of the host amalgam calculated in Sect. 2.2.2 ($k_{hos}$ in Eq. 1), as it takes its value from the linear regression of the empirical relationships of DB19 (see below). Expanding Eq. (A3) to the first-order term in the volume fraction of iron oxides results in the final analytic expression of the accretion IRI:

$$k_{acc} = k_0 + \frac{(3/2)I - k_0(n_{iox} - n_{hos})}{n_{hos}} v_{iox} = k_0 + l_1 v_{iox}. \tag{A4}$$

We assume now that dust particles of DB19 are two-component accretions whose IRI follows the linearized MG rule given by Eq. (A4), which we directly match to a linear regression of DB19 data to estimate the IRI of iron oxides. Given the non-trivial expression of the factor $I$ in Eq. (A4) (i.e., Eqs. A1 and A2), we apply an inversion procedure consisting of the following steps:

1. We perform a linear regression of the IRI retrievals of DB19 for all the measurement wavelengths with respect to the volume fractions of iron oxides in their dust aerosol samples, which we derive from the mass fractions using the mineral mass densities in Table 1. We thus estimate the empirical fit intercept ($l_0^*$) and slope ($l_1^*$).

2. We calculate the factor $I$ for a high number of input IRIs of iron oxides and then the analytical slope ($l_1$) in Eq. (A4) using the known RRIs of the host amalgam

(Sect. 2.2.2) and iron oxides (taken from SZ15) along with the host medium IRI as the empirical intercept: $k_0 = l_0^*$. We span 10 000 logarithmically equidistant IRI values between 0.001 and 1 for iron oxides, resulting in a relative accuracy of $\sim 0.07\%$.

3. We infer the IRI of iron oxides by selecting the input value that makes the analytical and empirical slopes equal: $l_1 = l_1^*$. We also derive the uncertainty in the inferred IRI by perturbing the empirical slope based on the fit error (varying the empirical intercept has a negligible impact).

Figure 4 shows our calculated IRI of iron oxides at VIS wavelengths, which is roughly intermediate between the IRIs for pure hematite and goethite measured by Bedidi and Cervelle (1993).

This derivation contains an inconsistency compared to our assumptions elsewhere in this work. The linear regression of the IRI retrievals from DB19, along with the unconstrained fit intercept used for the host medium, differs from the polynomial representation of Fig. 3. To calculate the IRI of accretions (Sect. 2.2.3), we choose the polynomial function to reduce the systematic low bias of the linear function that results for large mass fractions of iron oxides while the host amalgam (Sect. 2.2.2) is set in the limit of vanishing iron oxides. This suggests that the candidate mixing rules of Fig. A1 are not suitable for our IRI calculations for accretions and the host mixture, which are consistent with DB19 and SZ15. Nonetheless, deriving a mixing rule that diverges from linearity for small volume fractions of iron oxides while being consistent with Maxwell's equations is outside the scope of this work. In any case, this inconsistency has little practical impact given the small radiative effect of pure iron oxides in our model calculations (Fig. 10) but could be addressed in future studies.

## Appendix B: Calculation of dust optical properties

In this section, we describe the calculation of dust optical properties within the discrete SW bands of ModelE2.1 that we use in our experiments, including the HOM control run in which we assume globally homogeneous composition along with the EXT and INT experiments in which instead we implement a dependency upon the varying mineral composition.

### B1 Sets of complex refractive index

Given the CRIs for the mineral components at the seven wavelengths of DB19 (Sects. 2.2.2 and 2.2.3 and Appendix A), to improve the accuracy of our optical calculations, we extrapolate the CRIs to 20 wavelengths ranging from 0.30 to 1 µm which are a subset covering the UV–VIS band (0.30–0.77 µm) of the full set of SW wavelengths at

**Table B1.** Real and imaginary refractive indices ($n+ik$), for the subset of 20 default shortwave wavelengths ($\lambda$), used in this work for default dust (subscript "hom") in the control run with homogeneous composition (HOM) (Sinyuk et al., 2003; Volz, 1973), free crystalline iron oxides (subscript "iox"), the host amalgam (subscript "hos") and static accretion (subscript "acc") in the mineral experiment with external mixing configuration (EXT) (Di Biagio et al., 2019; Scanza et al., 2015). The best-fit parameter defining the polynomial fitting function ($p_1$ in Eq. 1), which is also used in the mineral experiment with internal mixing configuration (INT), and the CRI of default dust for the full set of shortwave wavelengths up to 4.114 μm are also reported.

| $\lambda$ (μm) | $n_{\mathrm{hom}}+ik_{\mathrm{hom}}$ | $n_{\mathrm{iox}}+ik_{\mathrm{iox}}$ | $n_{\mathrm{hos}}+ik_{\mathrm{hos}}$ | $n_{\mathrm{acc}}+ik_{\mathrm{acc}}$ | $p_1$ |
|---|---|---|---|---|---|
| 0.300 | $1.600+i8.700\times10^{-3}$ | $1.682+i2.871\times10^{-2}$ | $1.531+i1.352\times10^{-3}$ | $1.535+i6.832\times10^{-3}$ | $3.444\times10^{+1}$ |
| 0.325 | $1.595+i7.000\times10^{-3}$ | $1.814+i1.377\times10^{-1}$ | $1.522+i1.310\times10^{-3}$ | $1.530+i6.470\times10^{-3}$ | $3.389\times10^{+1}$ |
| 0.350 | $1.590+i5.800\times10^{-3}$ | $1.988+i2.175\times10^{-1}$ | $1.515+i1.247\times10^{-3}$ | $1.528+i6.202\times10^{-3}$ | $3.405\times10^{+1}$ |
| 0.375 | $1.585+i4.600\times10^{-3}$ | $2.187+i2.708\times10^{-1}$ | $1.511+i1.167\times10^{-3}$ | $1.529+i5.973\times10^{-3}$ | $3.473\times10^{+1}$ |
| 0.400 | $1.580+i3.600\times10^{-3}$ | $2.396+i3.004\times10^{-1}$ | $1.508+i1.074\times10^{-3}$ | $1.532+i5.731\times10^{-3}$ | $3.575\times10^{+1}$ |
| 0.425 | $1.576+i2.900\times10^{-3}$ | $2.601+i3.091\times10^{-1}$ | $1.506+i9.705\times10^{-4}$ | $1.535+i5.430\times10^{-3}$ | $3.691\times10^{+1}$ |
| 0.450 | $1.572+i2.400\times10^{-3}$ | $2.786+i2.995\times10^{-1}$ | $1.505+i8.607\times10^{-4}$ | $1.539+i5.037\times10^{-3}$ | $3.802\times10^{+1}$ |
| 0.475 | $1.568+i2.000\times10^{-3}$ | $2.936+i2.744\times10^{-1}$ | $1.504+i7.481\times10^{-4}$ | $1.542+i4.534\times10^{-3}$ | $3.889\times10^{+1}$ |
| 0.500 | $1.564+i1.800\times10^{-3}$ | $3.036+i2.366\times10^{-1}$ | $1.503+i6.362\times10^{-4}$ | $1.544+i3.924\times10^{-3}$ | $3.933\times10^{+1}$ |
| 0.525 | $1.561+i1.600\times10^{-3}$ | $3.071+i1.888\times10^{-1}$ | $1.502+i5.286\times10^{-4}$ | $1.544+i3.238\times10^{-3}$ | $3.916\times10^{+1}$ |
| 0.550 | $1.558+i1.400\times10^{-3}$ | $3.041+i1.369\times10^{-1}$ | $1.501+i4.330\times10^{-4}$ | $1.542+i2.550\times10^{-3}$ | $3.817\times10^{+1}$ |
| 0.575 | $1.555+i1.200\times10^{-3}$ | $2.978+i9.386\times10^{-2}$ | $1.499+i3.660\times10^{-4}$ | $1.539+i1.989\times10^{-3}$ | $3.619\times10^{+1}$ |
| 0.600 | $1.553+i1.100\times10^{-3}$ | $2.917+i7.320\times10^{-2}$ | $1.498+i3.450\times10^{-4}$ | $1.536+i1.645\times10^{-3}$ | $3.304\times10^{+1}$ |
| 0.625 | $1.551+i9.000\times10^{-4}$ | $2.877+i7.711\times10^{-2}$ | $1.496+i3.694\times10^{-4}$ | $1.533+i1.480\times10^{-3}$ | $2.898\times10^{+1}$ |
| 0.650 | $1.550+i8.000\times10^{-4}$ | $2.849+i9.299\times10^{-2}$ | $1.495+i4.149\times10^{-4}$ | $1.531+i1.385\times10^{-3}$ | $2.488\times10^{+1}$ |
| 0.675 | $1.548+i7.000\times10^{-4}$ | $2.828+i1.078\times10^{-1}$ | $1.494+i4.569\times10^{-4}$ | $1.530+i1.312\times10^{-3}$ | $2.158\times10^{+1}$ |
| 0.700 | $1.544+i7.000\times10^{-4}$ | $2.808+i1.163\times10^{-1}$ | $1.493+i4.857\times10^{-4}$ | $1.528+i1.257\times10^{-3}$ | $1.936\times10^{+1}$ |
| 0.800 | $1.540+i8.000\times10^{-4}$ | $2.744+i1.073\times10^{-1}$ | $1.492+i5.125\times10^{-4}$ | $1.526+i1.195\times10^{-3}$ | $1.716\times10^{+1}$ |
| 0.900 | $1.535+i1.000\times10^{-3}$ | $2.704+i7.996\times10^{-2}$ | $1.492+i5.086\times10^{-4}$ | $1.524+i1.115\times10^{-3}$ | $1.587\times10^{+1}$ |
| 1.000 | $1.530+i2.000\times10^{-3}$ | $2.683+i1.016\times10^{-1}$ | $1.491+i6.218\times10^{-4}$ | $1.520+i7.008\times10^{-4}$ | $2.392\times10^{+0}$ |
| 1.250 | $1.520+i3.000\times10^{-3}$ | | | | |
| 1.500 | $1.510+i5.000\times10^{-3}$ | | | | |
| 2.000 | $1.500+i8.000\times10^{-3}$ | | | | |
| 2.507 | $1.468+i9.856\times10^{-3}$ | | | | |
| 2.614 | $1.475+i1.139\times10^{-2}$ | | | | |
| 2.829 | $1.481+i4.194\times10^{-2}$ | | | | |
| 2.957 | $1.481+i4.194\times10^{-2}$ | | | | |
| 3.043 | $1.481+i2.920\times10^{-2}$ | | | | |
| 3.257 | $1.519+i1.759\times10^{-2}$ | | | | |
| 3.471 | $1.475+i1.317\times10^{-2}$ | | | | |
| 3.686 | $1.481+i1.075\times10^{-2}$ | | | | |
| 3.900 | $1.481+i6.383\times10^{-3}$ | | | | |
| 4.114 | $1.481+i4.444\times10^{-3}$ | | | | |

which the default dust CRI is prescribed in ModelE2.1 (Table B1). (The remaining five NIR bands cover wavelengths from 0.77 to 4 μm.) We use spline interpolation to spectrally extend the CRIs of host mixture and iron oxides, together with the fitting polynomial function, and then calculate the CRI for static and dynamic accretions using Eqs. (1) and (2) (Table B1).

## B2 Optical properties for default dust

We pre-calculate offline the optical properties of default dust used by the SW radiation scheme of ModelE2.1, which are the extinction and scattering efficiencies, along with the asymmetry parameter. First, we calculate monodisperse optical properties for spherical particles using the Mie code from Mishchenko et al. (2002) as functions of the default CRI (Table B1), which is globally uniform but varies across wavelengths, and the size parameter that is defined as $x = \frac{\pi d}{\lambda}$, where $d$ is the particle diameter and $\lambda$ the wavelength. We consider a size parameter grid of 5000 values ranging from 0.02 to 1000 with a constant logarithmic step. Next, we integrate the monodisperse optical properties over each aerosol size bin for the full set of default SW wavelengths (up to 4.114 μm; Table B1). For size integration, we use a step number distribution ($f_{\mathrm{N}}$) corresponding to a constant volume size distribution on the logarithmic scale within each bin and zero outside (Tegen and Fung, 1994):

$$f_{\mathrm{N}}(d) = \begin{cases} Cd^{-4} & \text{for} \quad d \in (d_{\mathrm{A}}, d_{\mathrm{B}}) \\ 0 & \text{for} \quad d \notin (d_{\mathrm{A}}, d_{\mathrm{B}}) \end{cases}, \tag{B1}$$

**Table B2.** Diameter bounds ($d_A$, $d_B$) of the five transported size bins of ModelE2.1, with the clay bin split into four sub-bins to allow more accurate calculation of the dust–radiation interaction at ultraviolet and visible wavelengths. The effective diameter ($d_{eff}$) for each bin is also reported.

| Bin | ($d_A$, $d_B$) (µm) | $d_{eff}$ (µm) |
|---|---|---|
| Clay | (0.1, 0.2) | 0.139 |
| | (0.2, 0.5) | 0.305 |
| | (0.5, 1.0) | 0.693 |
| | (1.0, 2.0) | 1.386 |
| Silt-1 | (2.0, 4.0) | 2.773 |
| Silt-2 | (4.0, 8.0) | 5.546 |
| Silt-3 | (8.0, 16.0) | 11.09 |
| Silt-4 | (16.0, 32.0) | 22.18 |

where $C$ is a normalization constant and ($d_A$, $d_B$) are the prescribed diameter bounds of the size bins reported in Table B2 (Perlwitz et al., 2015a). Equation (B1) is also used to calculate the effective diameter (twice the mean radius over the cross-sectional area distribution) corresponding to each bin (Table B2). The size-integrated single-wavelength optical properties are mapped onto a 10 nm resolution wavelength grid and are then averaged over the six SW bands weighted by the solar flux (Thekaekara, 1974).

## B3   Optical properties for mineral components

The optical properties for mineral components are calculated during runtime only in the UV–VIS band of ModelE2.1 (0.30–0.77 µm). After defining the CRI for each mineral component (Table B1), either globally uniform for free iron oxides, host mixture and static accretion (EXT) or spatially and temporally varying for dynamic accretion (INT), we pick intensive optical properties (extinction and scattering efficiency, asymmetry parameter) from an external look-up table that we pre-calculate using a grid of RRIs and IRIs specifically selected to cover typical dust values with high resolution (Table B3). The look-up intensive properties are already integrated over the eight size bins of ModelE2.1 (Table B2) for the subset of 20 default shortwave wavelengths through the same procedure described in Appendix B2. We then derive the optical properties for the actual CRI through bilinear interpolation of the look-up properties corresponding to the four closest grid CRIs. Finally, we average the single-wavelength optical properties over the UV–VIS band of ModelE2.1 following the same method used for default dust (Appendix B2).

**Code availability.** The version of ModelE used in this work, including the model source code and necessary configuration and input files, is available at Zenodo (Obiso et al., 2024, https://doi.org/10.5281/zenodo.10808381).

The Mie code (based on Mishchenko et al., 2002) for calculating the look-up tables of dust optical properties used in this work is publicly available at: https://www.giss.nasa.gov/staff/mmishchenko/Lorenz-Mie.html (last access: 12 March 2024).

**Data availability.** The analysis scripts for generating the figures of this work, along with the required input data and the instructions for the user, are available at Zenodo (Obiso et al., 2024, https://doi.org/10.5281/zenodo.10808381).

The output files of the three model experiments of this work are also available at https://portal.nccs.nasa.gov/datashare/giss-publish/pub/Obiso_ACP_2024 (NCCS, 2024) TS5 as stand-alone files, for users who are only interested in quickly accessing the model output without downloading the full analysis package.

The AERONET Version 3 inversion data product can be downloaded from https://aeronet.gsfc.nasa.gov/new_web/download_all_v3_inversions.html (Goddard Space Flight Center, 2024).

**Supplement.** The supplement related to this article is available online at: https://doi.org/10.5194/acp-24-1-2024-supplement.

**Author contributions.** VO and RLM, in collaboration with CPGP and MGA, designed the strategy for the coupling between minerals and radiation in ModelE2.1 and defined the methods, analysis and objectives of the experiments. VO implemented the coupling schemes in the model, ran the simulations, performed the analyses and wrote the manuscript. RLM supervised all the phases of the project. MGA helped to define the filtering conditions for selecting dusty scenes and generated the AERONET climatologies used for model evaluation, while GLS offered guidance about the AERONET product. CDB and PF gave advice on the interpretation of the retrieved imaginary indices and measured mineral fractions from natural dust aerosol samples. JPP, SEB and KT offered technical support for ModelE2.1 and contributed to identifying the best implementation strategies. All the co-authors reviewed the manuscript and provided comments and suggestions that improved several aspects of the project.

**Competing interests.** At least one of the co-authors is a member of the editorial board of *Atmospheric Chemistry and Physics*. The peer-review process was guided by an independent editor, and the authors also have no other competing interests to declare.

**Disclaimer.** Publisher's note: Copernicus Publications remains neutral with regard to jurisdictional claims made in the text, published maps, institutional affiliations, or any other geographical representation in this paper. While Copernicus Publications makes every effort to include appropriate place names, the final responsibility lies with the authors.

**Acknowledgements.** We thank the editor and the two anonymous reviewers, whose comments improved the manuscript.

**Table B3.** Grid of real ($n_{grd}$) and imaginary ($k_{grd}$) refractive indices included in the external look-up table of size-integrated single-wavelength intensive optical properties (extinction and scattering efficiency, asymmetry parameter) used for the calculation of optical properties for mineral components.

| $n_{grd}$ |
|---|
| 1.490–1.500–1.510–1.520–1.530–1.540–1.550–1.560 |
| 1.570–1.580–1.590–1.600–2.000–2.400–2.800–3.200 |

| $k_{grd}$ |
|---|
| 0–0.0001–0.0002–0.0003–0.0004–0.0005–0.0006–0.0007–0.0008–0.0009 |
| 0.001–0.002–0.003–0.004–0.005–0.006–0.007–0.008–0.009 |
| 0.01–0.02–0.03–0.04–0.05–0.06–0.07–0.08–0.09 |
| 0.1–0.2–0.3–0.4–0.5–0.6–0.7–0.8–0.9–1–1.1–1.2–1.3 |

Computational resources supporting this work were provided by the NASA High-End Computing (HEC) program through the NASA Center for Climate Simulation (NCCS) at Goddard Space Flight Center.

We thank Reto A. Ruedy for his valuable technical support with ModelE and the supercomputer infrastructure.

We are grateful to Michael I. Mishchenko, whose vast legacy includes publicly available codes for calculating the scattering of radiation by aerosols.

We also thank the AERONET scientists and staff for establishing the stations, maintaining the instruments and making the data publicly available.

Claudia Di Biagio and Paola Formenti acknowledge the AERIS data center (https://www.aeris-data.fr; last access: 19 February 2024) of the DATA TERRA French national research infrastructure for distributing and curating the data produced by the CESAM chamber through the hosting of the EUROCHAMP data center (https://data.eurochamp.org; last access: 19 February 2024).

**Financial support.** Vincenzo Obiso was supported by the NASA Postdoctoral Program at the NASA Goddard Institute for Space Studies, administered by Oak Ridge Associated Universities under contract with NASA. Additional support was provided by the NASA Modeling, Analysis and Prediction Program (NNG14HH42I) and the Earth Surface Mineral Dust Source Investigation (EMIT), a NASA Earth Ventures Instrument (EVI-4) mission, both of which also supported Ron L. Miller.

Carlos Pérez García-Pando, María Gonçalves Ageitos and Vincenzo Obiso acknowledge funding by the European Research Council under the Horizon 2020 research and innovation program through the ERC Consolidator Grant FRAGMENT (grant agreement no. 773051), the AXA Research Fund through the AXA Chair on Sand and Dust Storms at the Barcelona Supercomputing Center (BSC), the European Union's Horizon 2020 research and innovation program under grant agreement no. 821205 (FORCeS) and the national grants HEAVY PID2022-140365OB-I00 and BIOTA PID2022-139362OB-I00 funded by MICIU/AEI/10.13039/501100011033 and by ERDF and the EU.

Jan P. Perlwitz has been funded by NASA grant nos. 80NSSC19K0056 (Modeling, Analysis and Prediction Program), 80NSSC19K0984 (Atmospheric Composition Modeling and Analysis Program) and 80NSSC22K1663 (Goddard Institute for Space Studies Model Development).

Gregory L. Schuster is supported by the National Aeronautics and Space Administration through the Science Mission Directorate's Earth Science Division.

The laboratory experiments to retrieve the dust refractive indices in Di Biagio et al. (2017, 2019) that feed this work were supported by the French national program LEFE/INSU (Les Enveloppes Fluides et l'Environnement/Institut National des Sciences de l'Univers) and by the OSU-EFLUVE (Observatoire des Sciences de l'Univers-Enveloppes Fluides de la Ville à l'Exobiologie) through dedicated research funding to the RED-DUST project. They were acquired with the CESAM simulation chamber, a national facility of the CNRS-INSU (Institut National des Sciences de l'Univers of the Centre National de la Recherche Scientifique) and part of the French ACTRIS research infrastructure. The CESAM simulation chamber also received funding from the European Union's Horizon 2020 research and innovation program through the EUROCHAMP-2020 Infrastructure Activity under grant agreement no. 730997. Claudia Di Biagio was supported by the Centre National des Etudes Spatiales (CNES) and by the CNRS via the Labex L-IPSL, which is funded by the ANR (grant no. ANR-10-LABX-0018).

**Review statement.** This paper was edited by Manvendra Krishna Dubey and reviewed by two anonymous referees.

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

## Remarks from the language copy-editor

CE1    Please note that we do not permit colons to separate a preposition from the rest of a sentence.

CE2    Please note that we do not permit colons to separate a verb from the rest of a sentence.

## Remarks from the typesetter

TS1    Please note that the current supplement link is a placeholder that will be substituted during the publication.

TS2    Deleting these parts in the equation is a bigger impact and needs the approval from the editor. Please write a statement why this change needs to be done. Thank you.

TS3    Please note the verb is correct, as you cite two panels.

TS4    No, it is not our standard to write an equation on two lines. Please confirm the fraction.

TS5    Please note that every URL cited in this section has to be cited in the reference list. Please check all my suggestions and if necessary, please correct.

TS6    Please confirm.

TS7    Please confirm.

TS8    Please note that place is cited before the date of the symposium.