# Peer review of "Observationally constrained regional variations of shortwave absorption by iron oxides emphasize the cooling effect of dust"

_EGUsphere, 2023_

## Author Response (AR1)

**Responses to Reviewers**

RC1:

Overall, this paper presents a commendable piece of work. It provides valuable insights into dust optical properties and modeling on a global scale. However, there are a few areas where further clarification is required.

We thank the reviewer for the helpful comments which contributed to improving our current manuscript by identifying unclear parts, and provided very useful suggestions for future research.

Line 125: 'One half of dust particles are assumed to be soluble.' Is it typical to assume that half of the dust particles are soluble in your simulation? Could you please provide a justification for this assumption? Additionally, are you employing a kappa-Kohler framework to determine the solubility?

Dust particles are largely insoluble at the time of emission, but during atmospheric transport their solubility increases through heterogeneous uptake of trace gases, such as $SO_2$ and $HNO_3$, and the formation of soluble sulfate or nitrate coatings on the particle surface (e.g. Usher et al., 2003). For simplicity and computational efficiency, dust solubility is not explicitly calculated in our current study. Previously, Bauer and Koch (2005) found that a constant solubility of 0.5 yielded a similar global dust load compared to a model version where the formation of sulfate coatings on the dust particles was explicitly calculated. Relying on these results, we also assume a globally uniform solubility of 0.5 for all minerals in our study. We defer an explicit calculation of heterogeneous uptake, particle solubility, and its dependence upon aerosol mineral composition to a future study.

To better explain our assumption, we added the following paragraph in the manuscript (Section 2.1; Lines 113 – 118):

"*Dust particles are largely insoluble at emission, but during atmospheric transport their solubility increases through the formation of sulfate or nitrate coatings on the particle surface (e.g. Usher et al., 2003). Bauer and Koch (2005) found that a constant solubility of 0.5 yielded a similar global dust load compared to a model version where the formation of sulfate coatings on dust particles was explicitly calculated. Relying on these results, we also assume a globally uniform dust solubility of 0.5; however, we do not account for the hygroscopic growth of the dust particles in the radiative calculations.*"

Additional References:

Bauer, S. E. and Koch, D. (2005). Impact of heterogeneous sulfate formation at mineral dust surfaces on aerosol loads and radiative forcing in the Goddard Institute for Space Studies general circulation model. *J. Geophys. Res.*, 110, D17202.

Usher, C. R., Michel, A. E. and Grassian, V. H. (2003). Reactions on mineral dust. *Chem. Rev. 2003*, 103, 4883 − 4940.

Line 127: Could you elaborate on the similarity between your un-calibrated emissions and the observed concentrations? Specifically, what is the magnitude of the calibration factor? How much larger is it compared to the un-calibrated emissions?

Our discussion of calibration should have been more specific. There are two parts to our calibration: the global mass of emitted dust and its distribution across particle diameters. The distribution of

emitted sizes is an unsettled topic of research. We specify the normalized size distribution using a combination of a theoretical relation by Kok (2011) and measurements of dust concentration during a dust outbreak by Kandler et al. (2009), assuming that the distribution is independent of wind speed, as described in Perlwitz et al. (2015a).

The coefficient linking the emitted dust mass to wind speed at a specific location can be derived empirically by measuring wind and concentration. However, the wind provided by an Earth System Model is an average over a grid box, whose dimensions are in this case on the order of 200 km. Thus, the wind at the specific location needed to calculate dust emission is not provided by the model, and another strategy needs to be used to relate the mass of dust emitted from a single grid box to the grid-box average wind. One approach is to adjust or 'calibrate' the relation between grid-box wind and emission using a single global factor so that the resulting global DOD lies within the observed range. In this work, we use a single global calibration factor that results in a global-mean DOD equal to roughly 0.02 (averaged over the UV-VIS band). This is at the lower end of the range 0.02 – 0.04 estimated by Ridley et al. (2016), based upon observations supplemented by model output. In general, the calibration factor is expected to depend on model resolution and the PBL parameterization, along with other physical processes represented by the model. Moreover, there is really no such thing as an 'un-calibrated' model. Every model must specify a factor coupling wind speed to emitted mass where both are averaged over a model grid box. For these reasons, we do not provide the calibration factor because we feel that its precise value is not meaningful to other models (including other versions of the GISS model). In this study, an important aspect is that we use the same calibration factor in all experiments, so that differences in the global mass load among the three experiments are attributable only to a different radiative feedback on emission.

In the revised version of the manuscript, we expanded our explanation of calibration as follows (Section 2.1; Lines 99 – 106):

"*The global magnitude of dust emission is uncertain due to a scarcity of direct observations and the limited spatial and temporal resolution of the model wind speed that has a non-linear influence upon emission. Typically, models adjust the relation between grid-box wind and emitted mass using a globally invariant calibration factor so that the model is in optimal agreement with a range of observations (e.g Cakmur et al., 2006). Here, our calibrated emission yields a global model DOD near 0.02 that is within the observationally constrained range estimated by Ridley et al. (2016) (Section 3.1.1). Our global calibration factor is identical in all our experiments, so that differences in DOD and mass load among the experiments are the result of a varying radiative feedback upon the surface wind speed and emission (e.g. Miller et al., 2004; Pérez et al., 2006; Heinold et al., 2007; Ahn et al., 2007).*"

Line 195-205: Regarding the refractive index mixing rules, I appreciate the decision to exclude the MG or BG methods. While the volume weight mixing rule seems acceptable, I believe the Lorentz-Lorenz mixing rule is more suitable for aerosols. This rule has been found to be particularly effective in situations where the refractive indices of individual components in the mixture are closely aligned. For further information on mixing rules for aerosols, I recommend referring to the paper by Liu.
Liu, Y., & Daum, P. H. (2008). Relationship of refractive index to mass density and self-consistency of mixing rules for multicomponent mixtures like ambient aerosols. Journal of Aerosol Science, 39(11), 974–986. https://doi.org/10.1016/j.jaerosci.2008.06.006

We appreciate the recommendation of this paper (Liu and Daum, 2008) as it contains theoretical information for calculating the effective refractive index of a composite material that is consistent with the Lorentz-Lorenz equation. The derivation of an empirical relation between the refractive index and the mass density of a material is also interesting, as it allows circumventing the lack of

measurements of effective molecular weight and polarizability for atmospheric aerosols. Assuming, as a first approximation, that atmospheric aerosols can be considered as Lorentz-Lorenz mixtures (which is one of the uncertainties discussed in Section 4 of Liu and Daum, 2008), and assuming that the Lorentz-Lorenz relation holds for the complex refractive index (in that study, the authors only focus on the real index), we will definitely investigate the potential application of the Lorentz-Lorenz mixing rule for dust minerals in future studies. Regarding the host mixture in this current work, which is the specific case referred to by the reviewer, Liu and Daum (2008) also demonstrate that the Lorentz-Lorenz mixing rule reduces to the linear volume-mean mixing rule for quasi-homogeneous mixtures (Section 3.3). In a first approximation, this condition applies to our host mixture, as host minerals have relatively similar refractive indices. Therefore, even though the Lorentz-Lorenz mixing rule represents a more general method, the volume-mean mixing rule seems to be an acceptable approximation for our host mixture (as highlighted also in reviewer's comment).

Based on reviewer's suggestion, we better justified the use of the volume-mean mixing rule for our host mixture as follows (Section 2.2.2; Lines 212 – 218):

"*Given these limitations, we calculate the CRI of the host mixture using the VM (volume-weighted mean) rule, which prescribes the real and imaginary parts of the composite CRI in proportion to the volume fractions of each mineral. The VM rule can be derived as a linear approximation to the Lorentz-Lorenz mixing rule for the case of a quasi-homogeneous mixture of different components with similar refractive indices (Liu and Daum, 2008). In a first approximation, this condition applies to our host mixture that is composed of minerals with relatively similar refractive indices. While VM predicts greater absorption compared to MG or BG when the inclusion is highly absorbing, this bias is less important for the amalgam of host minerals whose absorption is small compared to iron oxides.*"

Figure 2: I thought it was interesting that the complex refractive indexes are close to those of salts, measured in:
Bain, A., Rafferty, A., & Preston, T. C. (2019). The Wavelength-Dependent Complex Refractive Index of Hygroscopic Aerosol Particles and Other Aqueous Media: An Effective Oscillator Model. Geophysical Research Letters, 46(17–18), 10636–10645. https://doi.org/10.1029/2019GL084568

This is an interesting comparison. In Figure 3 of Bain et al. (2019), we see that all the considered salts have an imaginary index close to $10^{-8}$ for wavelengths up to 0.7 µm. These values are very much consistent with those of quartz, which is the least absorbing mineral (based on Scanza et al., 2015).

Line 330-335: Considering that the dust emissions have been calibrated (as stated in Line 127), it is indeed anticipated that there will be an agreement with the observations. However, please correct me if I have misunderstood this section.

The observationally constrained literature estimates of the global dust mass that we use for model evaluation (e.g. Adebiyi and Kok, 2020) are independent of the measurements used to calibrate emission with one exception: the concentration measurements by Kandler et al. (2009) that we use to specify the normalized size distribution of emitted dust are one of 27 distinct measurements used by Adebiyi and Kok (2020) to estimate the normalized size distribution of global dust mass. We believe that this commonality makes only a small contribution to our model agreement with the global mass of coarse dust (with particle diameters of 5 – 20 µm) estimated by Adebiyi and Kok (2020). Note that we compare the absolute mass of coarse dust, which depends additionally on the global dust load and not just on the emitted fraction of coarse particles that we specify.

As mentioned in a previous comment, our calibration of dust emission results in a global DOD near 0.02, which is at the lower end of the range 0.02 – 0.04 estimated by Ridley et al. (2016). Despite our probable underestimate of DOD, our model produces $PM_{20}$ that is on the high end of a semi-observational constraint (Kok et al., 2017), which is based on the same DOD (0.03 as the central value) estimated by Ridley et al. (2016). Given that the dust load in the fine size range (0.1 – 5 µm in diameter) is similar in our model and in Kok et al. (2017), we suggest that this discrepancy is partly due to our representation of dust particles as spheres in the radiative calculations, which typically have a lower extinction efficiency than non-spherical particles (Huang et al., 2023; Kok et al., 2017).

RC2:

This study investigates the effect of heterogeneous mineral composition on dust aerosol optical properties especially absorption, and the impact on estimated dust direct radiative effect (DRE) in the shortwave. Two approaches are implemented to account for the mixing of absorbing iron oxides and non-absorbing dust minerals, compared to the assumption of the globally uniform composition. It is found that the consideration of spatially and temporally dependent mineral composition in dust leads to better agreement with the AERONET retrieved single scattering albedo and generally results in a larger cooling of dust than the homogeneous assumption. The results of this study demonstrate the importance of accounting for the variability in dust mineralogy in the calculation of dust forcing and showed the sensitivity to the mixing assumptions. These insights are very useful for improving the global modeling of dust and understanding the differences between the models and from the observations. I have a few general comments and some specific comments are given below that need further clarification.

We sincerely appreciate the valuable comments and suggestions of the reviewer, which allowed us to clarify some aspects of our methods and results, and contributed to making our manuscript more robust.

The manuscript discussed the model results of dust optical properties and radiative effects with different approaches first and then evaluated the calculated dust optics with AERONET. I'd suggest moving the model evaluation up front before the discussions of the sensitivity studies. The comparison with the AERONET observations would add value to the discussion of dust DREs with different methods and show how the observationally constrained single scattering albedo affects the calculated direct effect.

We originally considered the reviewer's suggestion of moving the comparison with AERONET observations before the sections presenting the modeling results and sensitivity tests. However, the current structure of the manuscript allows us to first present the main qualitative and quantitative features of the two mineral coupling schemes compared to the control case, which we then use to analyze the comparison with AERONET observations. While we recognize the advantages of the reviewer's suggestion, we prefer to maintain our current structure of the manuscript.

For the calculation of dust DRE, vertical distribution is also an important parameter, in addition to dust optical depth and single scattering albedo. Although the representation of dust mineral composition may not have a large impact, it should be discussed how the model representation of dust vertical distribution in this model may affect its estimated dust DRE especially when compared to other studies. Also, this should be mentioned in the discussion of "source of model uncertainty".

This is a good point. Even in clear-sky conditions, the altitude of an absorbing dust layer (with fixed DOD and SSA) affects the dust DRE at the TOA, especially over lowly-reflecting surfaces, mainly because it affects the dust absorption of solar radiation that is back-scattered by atmospheric molecules: the higher the dust layer the less negative the DRE at the TOA (Meloni et al., 2005). A varying absorption profile of dust may also impact the DRE at the TOA. For example, Noh et al. (2016) calculated that a layer of scattering aerosols above a layer of absorbing aerosols makes the DRE at TOA less positive than the opposite configuration, even keeping the same column average AOD and SSA. This because incoming SW radiation is back-scattered by the upper layer before being absorbed by the bottom layer (and vice versa). This applies to dust with varying composition, whose absorption properties are expected to vary along the column. Vertical distribution also influences the climate response, even for a fixed DRE (Miller and Tegen, 1999). In general, evaluation of the vertical

distribution of dust has received less attention by modelers, in part because of the limited observational constraints. In the future, we hope to evaluate against ATom and CALIOP.

We added two paragraphs about this uncertainty in our comparison with other models and their DRE, as suggested by the reviewer. First, in our evaluation of the homogeneous dust experiment (Section 3.1.1; Lines 382 – 387):

"*We highlight that the calculation of dust DRE at TOA also depends upon the vertical distribution of dust, in addition to optical properties like DOD and SSA. Even in clear-sky conditions, the altitude of a dust layer modulates its absorption of solar radiation that is back-scattered by the air molecules, especially over weakly-reflecting surfaces, so that a higher dust layer results in a less negative SW DRE at TOA even with a fixed column average DOD and SSA (Meloni et al., 2005). This introduces a poorly constrained uncertainty in the comparison among different model calculations, as evaluation of the dust vertical profile has typically received less attention in modeling studies.*"

Second, in our evaluation of the mineral experiments (Section 3.1.3; Lines 475 – 480):

"*Despite both these studies using the same soil mineralogy map from Claquin et al. (1999) that we use in this work, differences in the regional and vertical distribution of minerals and their optical properties may partly explain the contrasting results. For example, the absorption profile of dust that is determined by the vertical distribution of different minerals and sizes may affect the calculation of the dust SW DRE at TOA, even with a fixed column-integrated DOD and SSA. This is because the relative height of absorbing and scattering layers modulates the amount of radiation absorbed in the column (Noh et al., 2016).*"

Additional References:

Meloni, D., Di Sarra, A., Di Iorio, T. and Fiocco, G. (2005). Influence of the vertical profile of Saharan dust on the visible direct radiative forcing. *J. Quant. Spectrosc. Radiat. Transf.*, 93, 397 – 413.

Miller, R. L. and Tegen, I. (1999). Radiative forcing of a tropical direct circulation by soil dust aerosols. *J. Atmos. Sci.*, 56(14), 2403 – 2433.

Noh, Y. M., Lee, K., Kim, K., Shin, S.-K., Müller, D. and Shin, D. H. (2016). Influence of the vertical absorption profile of mixed Asian dust plumes on aerosol direct radiative forcing over East Asia. *Atmos. Environ.*, 138, 191 – 204.

This study examines two mixing rules regarding the absorbing and non-absorbing substances in dust. It would be interesting to compare the results from the DRE sensitivity studies, e.g., HOM vs EXT and HOM vs INT, with previous studies such as DB19, SZ15 and Li et al. 2021.

Following this good suggestion, we carried out a comparison with Scanza et al. (2015) and Li et al. (2021). Li et al. (2021) do not report results of simulations with homogeneous dust, so we took the baseline experiment with CAM5 using the soil map of Claquin et al. (1999) to compare their dust AOD, SSA and DRE to our values from the mineral experiments (EXT and INT). On the other hand, Scanza et al. (2015) do compare experiments with (CAM4-m and CAM5-m) and without (CAM4-t and CAM5-t) considering mineral speciation of dust.

We added the following paragraph to the manuscript (Section 3.1.3; Lines 466 – 475):

*"The increase of global cooling at TOA due to varying mineralogy (both in our EXT and INT experiments) partially contrasts with the results of SZ15. In that study, the reported dust SW DRE at TOA is less negative in the experiments with explicit mineralogy ($-0.04$ and $-0.08 W \cdot m^{-2}$ in CAM4 and CAM5, respectively) compared to the corresponding experiments with unspeciated dust ($-0.14$ and $-0.33 W \cdot m^{-2}$ in CAM4 and CAM5, respectively). This decrease of global cooling at TOA is associated with greater absorption indicated by a lower SSA in CAM5, but surprisingly with a higher SSA in CAM4. Using CAM5 with explicit mineralogy, Li et al. (2021) calculated a SW DRE at TOA of smaller magnitude ($-0.18 W \cdot m^{-2}$) compared to our values from the mineral experiments ($-0.30$ and $-0.34 W \cdot m^{-2}$ from EXT and INT, respectively; Table 3), despite their higher DOD of 0.03. The small magnitude of their DRE efficiency ($-5.94 W \cdot m^{-2}$) may be partly attributed to their quite low SSA of 0.89. (The latter, however, is inferred from the total aerosol SSA and thus possibly affected by non-dust absorbing species.)"*

Moreover, in our manuscript we already compare our DRE calculations with Di Biagio et al. (2020). Regarding Di Biagio et al. (2019), the authors report measurements and retrievals of composition, size and optical properties of several dust aerosol samples. However, they do not perform calculations of dust DRE. Moreover, their estimates of dust SSA are based on aerosolized soil samples collected from different source locations. Therefore, a possible comparison with our model SSA at the same locations would have limited value because the model calculates the SSA of dust in the column, which is affected by dust plumes transported at different altitudes and from different locations. The only remaining comparison would be between the mineral composition measured by Di Biagio et al. (2019) in their aerosolized soil samples and the soil mineral fractions prescribed in our model based on the soil map of Claquin et al. (1999). However, such comparison is difficult to interpret due to the confounding effects of biases in Claquin et al. (1999) along with the effect of differing sample locations and resolution. Moreover, this comparison would substantially be an evaluation of the soil map of Claquin et al. (1999), which has been recently performed by Gonçalves et al. (2023) but is out of the scope of our current study. For all these reasons, despite basing our modeling of mineral-radiation interaction on the work of Di Biagio et al. (2019), we have not included a comparison between our results and the results of Di Biagio et al. (2019) in our current work.

Specific comments:

Section 2.2.1: How does the global model determine the mass fraction of each dust mixture in EXT in a specific time and location: host mixture, static accretion and free iron oxides?

In all experiments, the model separately calculates the mass of 15 mineral species (comprised of 7 pure host minerals, 7 accretions of these host minerals with a 5% mass fraction of embedded iron oxides, and finally pure iron oxides) distributed across 5 size bins (clay plus 4 silt bins), by multiplying the dust emitted flux by the size-resolved mineral fractions at emission, and then transporting and depositing each mineral species during the simulation (Perlwitz et al., 2015a). Our different mixing configurations for minerals only apply to the interaction with radiation, not to the transport. Thus, in the EXT scheme mentioned by the reviewer, when the radiation routines are called at each time and location, we assign optical properties to each of the 15 mineral species according to its 'radiative' category: host mixture, static accretion or free iron oxides.

Line 262: how does the AeroTAU0 dataset differ from the standard Level 1.5 data products? This is a bit unclear.

The full set of quality assurance criteria for the Level 2 inversion product in AERONET (Table 3 of Holben et al., 2006) includes the condition: $AOD_{440} > 0.4$. The AERONET Level 1.5 database provides a flag identifying the measurements that meet all the Level 2 quality assurance criteria except

for the condition on the $AOD_{440}$. These data can therefore be considered 'quasi' Level 2 data, as they would just need to be filtered only for $AOD_{440} > 0.4$ to become Level 2 data. We use fully Level 2 data for the comparison of SSA, because the condition on $AOD_{440}$ is mostly important for retrieving SSA with a reasonable error. In contrast, we skip this condition for the comparison of dust AOD, as AOD is directly measured and not retrieved, aiming to increase the sample of AERONET data available for comparison to the model.

Line 292: how sensitive are the results to this criteria: K_rir<0.0042? Please add a couple of sentences to justify it.

The condition $IRI_{rir} < 0.0042$ comes from Schuster et al. (2016). In that study, the authors apply strict filters to AERONET retrievals of the IRI to separate 'pure' dust retrievals at West African AERONET sites (FVF below 0.05 and depolarization ratio above 0.2) from carbonaceous retrievals at South American sites during the peak season of biomass burning. They found that 95% of retrieved dust IRI at red-NIR wavelengths are below 0.0042 and 95% of biomass burning imaginary indices are above 0.0042 (see their Figure 5). In other words, this threshold allows an overlapping of only 5% of the two distributions. This result derives from the strong absorption by black carbon at long-VIS and NIR wavelengths. We apply the condition $IRI_{rir} < 0.0042$ as a third filter to select dusty scenes in AERONET, after the size filter (FVF < 0.10) and the SSA filter ($SSA_{675} > SSA_{440}$). This third condition aims to reduce the presence of absorbing carbonaceous aerosols in scenes that are not filtered out by the first two conditions, for example scenes dominated by dust in which small fractions of black carbon (either in the remaining fine mode or attached to coarse dust particles) are still able to affect the absorption of the mixture. The sensitivity of our results to the condition $IRI_{rir} < 0.0042$, therefore, depends on how many scenes like that are present in our sample. Given that the first two filters are already quite strict, we expect that removing this third condition would not have an extremely significant impact on our comparison with AERONET, also because we only consider monthly means with a minimum of 80 hourly retrievals. Using a higher threshold would admit some more retrievals into our sample, but with the risk of including more scenes contaminated by absorbing carbonaceous aerosols. Conversely, a lower threshold would potentially discard some scenes with uncontaminated dust.

We have not tested the sensitivity of our results to this filtering condition. Nonetheless, we added the following sentences in the manuscript to justify more clearly our choice for the threshold for $IRI_{rir}$ (Section 2.3; Lines 310 – 314):

"*The threshold of 0.0042 has been proposed by Schuster et al. (2016) to separate 'pure' dust retrievals at West African AERONET sites (identified by a FVF below 0.05 and a depolarization ratio above 0.2) from retrievals of 'pure' carbonaceous samples at South American sites during the peak season of biomass burning: the threshold value of 0.0042 allows an overlap of only 5% between the distributions of IRI at red-NIR wavelengths retrieved for dust and biomass burning species, with the former generally exhibiting a lower IRI.*"

Line 300: 80 hours per month is barely more than 3 days. For those sites/months, the monthly means calculated may not be comparable to the model monthly averages. This caveat should be mentioned in the comparison with AERONET.

This represents an intrinsic limitation of the comparison between monthly model output with AERONET retrievals. We tried different thresholds for the minimum of hourly measurements required to consider a month for a given station. Selecting dust scenes with our strict conditions already filters out a considerable amount of hourly data. Increasing the minimum threshold would further decrease the number of data. Thus, the used threshold of 80 hourly data per month is a

compromise between the regional-seasonal representativeness and that of the monthly means. However, the reviewer is correct that using as few as 80 hourly data for calculating a monthly mean represents an uncertainty in our comparison.

We added the following sentences in our discussion (Section 2.3; Lines 320 – 323):

"*Our chosen threshold for the minimum number of measurements in each month represents a compromise between minimizing the uncertainty of the monthly mean while maximizing the number of stations and months in the sample. However, the threshold contributes an intrinsic uncertainty to the comparison between the model monthly output and AERONET.*"

Section 2.3: AERONET has an AE product, which has been used to distinguish particles with large vs small sizes. Can you comment why not use this product, instead of using the FVF and absorption spectra?

The reviewer is correct that AE has been used widely in studies comparing model results with AERONET to distinguish large and small particles. Values of AE below 1 roughly indicate size distributions dominated by coarse mode aerosols, while AE above 2 are typically associated with fine mode aerosols. However, AE is not a measurement of size: it only describes the spectral dependence of the measured extinction that is approximated as linear in the log scale. The relationship between AE and size distribution is only qualitative and is not unique. Thus, any given AE can correspond to a large range of FVF and fine mode effective radii (e.g. Figure 11 of Schuster et al., 2006). On the other hand, the FVF retrieved in AERONET is superior to AE because it is constrained by the almucantar scan in addition to being constrained by the spectral dependence of extinction. For this reason, we use FVF in our filtering method as an indicator of size distribution that is more robust and precise than AE.

Additional References:

Schuster, G. L., Dubovik, O. and Holben, B. N. (2006). Angstrom exponent and bimodal aerosol size distributions. *J. Geophys. Res.*, 111, D07207.

Lines 341-342: it would be helpful to clarify how much the non-sphericity assumption would increase the particle scattering, e.g., based on the estimates from the previous studies such as Kok et al. (2017), compared to the low bias in DOD (by percent) in this study.

We considered this question, in part to understand why our global mass is at the high end of an observationally constrained range (Kok et al., 2017), while our total DOD is at the low end (Ridley et al., 2016). The literature has conflicting results about the importance of non-sphericity to optical properties typically used in models (e.g. Mishchenko et al., 1997; Räisänen et al., 2013; Kok et al., 2017; Huang et al., 2023). In a separate project with compositionally homogeneous particles, we find an enhancement of dust extinction by triaxial ellipsoids that is comparable to that found by Kok et al. (2017). We expect that ellipsoidal effects in the present study would significantly reduce the low bias of our DOD while keeping the good agreement of our size-distributed mass load with the estimate by Adebiyi and Kok (2020). However, one complication that prevents us from making any quantitative estimate is that, for linking dust absorption to varying composition, we use empirical relations between the dust IRI at solar wavelengths and the fractional iron oxide mass that is retrieved from measurements by Di Biagio et al. (2019) assuming spherical particles. Inferring non-spherical radiative impacts has to be done carefully to be consistent with the assumptions underlying the empirical relations, a topic that is beyond the current study.

Additional References:

Mishchenko, M. I., Travis, L. D., Kahn, R. A. and West, R. A. (1997). Modeling phase functions for dustlike tropospheric aerosols using a shape mixture of randomly oriented polydispersed spheroids. *J. Geophys. Res.*, 102(D14), 16831 – 16847.

Räisänen, P., Haapanala, P., Chung, C. E., Kahnert, M., Makkonen, R., Tonttila, J. and Nousiainen, T. (2013). Impact of dust particle non-sphericity on climate simulations. *Q. J. R. Meteorol. Soc.*, 139, 2222 – 2232.

Line 350: suggest to also include the mean forcing value from Kok et al (2017)

We thank the reviewer for this suggestion. In addition to the reported range of DRE values constrained by Kok et al. (2017), we added the mean value (Section 3.1.1; Line 373):

" ... *(from approximately $-0.80$ to $-0.15 W \cdot m^{-2}$, with a mean value near $-0.50 W \cdot m^{-2}$).*"

Line 341: "warming effect": is this referring to the SW effect of the coarsest particles? Should the SW effect be more negative for larger particles, which are more efficient in scattering (higher SSA that smaller dust particles)? Please clarify.

We assume this comment refers to line 351, where we compare our DRE with Kok et al. (2017). We find our DRE ($-0.25$ Wm$^{-2}$) to be closer to the less cooling end of the range constrained by Kok et al. (2017) (roughly from $-0.80$ to $-0.15$ Wm$^{-2}$), even after adjusting our DOD to their higher value of 0.03 ($-0.38$ Wm$^{-2}$). Typically, at solar wavelengths, with a non-zero imaginary index, coarse dust particles absorb more than fine particles (resulting in a lower SSA) because of their larger volume. In contrast, fine particles are more efficient scatterers because scattering efficiency peaks for sizes similar to the wavelength (Hansen and Travis, 1974). Our size distribution extends to 32 μm in diameter (with enhanced emission of coarse particles), while the DRE estimates of Kok et al. (2017) refer to diameters below 20 μm (with underrepresented mass load in the 5 – 20 μm coarse diameter range, according to Adebiyi and Kok, 2020). This means that our calculation of dust DRE accounts for higher fractions of warming coarse particles compared to Kok et al. (2017). Thus, we partly attribute to this size discrepancy our less negative adjusted DRE ($-0.38$ Wm$^{-2}$) compared to the central value (~ $-0.50$ Wm$^{-2}$ for the SW) from Kok et al. (2017).

We added a clarifying sentence in the revised manuscript (Section 3.1.1; Lines 373 – 375):

"*This may partly be due to the warming effect of our higher fraction of absorbing coarse particles (Figure 7-B), which are underrepresented by Kok et al. (2017) in the diameter range of $5-20μm$ (according to Adebiyi and Kok, 2020) or not included at all above 20μm.*"

Line 356: same question as on line 341.

In this case, we compare with Di Biagio et al. (2020) who considered giant particles with diameters up to 150 μm (indicated by aircraft and field measurements), which are not included in our calculation. That may partly explain their less negative DRE efficiency ($-9.62$ Wm$^{-2}$ per unit DOD at 0.550 μm) compared to ours ($-12.75$ Wm$^{-2}$ per unit DOD in the UV-VIS band) for the same reason explained in the previous comment (i.e. large particles warm more).

We better clarified this mechanism also here (Section 3.1.1; Lines 378 – 381):

*"Di Biagio et al. (2020) extended the emitted PSD to diameters above 20µm (up to 150µm) by fitting airborne measurements from the FENNEC campaign (Ryder et al., 2013a, b). The warming effect of their giant particles, whose diameters are not included in our model, may partly explain their less negative DRE efficiency."*

Line 394: why is the increase of dust scattering relative to the imaginary part of refractive index? It became clear after reading the rest of the paragraph. But this sentence is confusing. Suggest rephrasing.

We thank the reviewer for identifying this unclear sentence. The meaning of this specific sentence is that we find a general increase of dust scattering due to varying composition if we take the IRI of Sinyuk et al. (2003) as the reference (HOM experiment). If we had used a lower imaginary index in the HOM experiment, likely we would have obtained scattering increases in some areas and decreases in other areas in the mineral experiments.

We rephrased the sentence in a clearer way (Section 3.1.2; Lines 422 – 426):

*"It is important to point out that, while the emphasized regional differences in SSA are a direct consequence of the varying mineral composition, the general increase of dust scattering depends upon the specific IRI that we use for unspeciated dust in the control experiment. The IRI retrieved by Sinyuk et al. (2003) is just one of the possible IRI sets for dust available in literature, and in general is less absorbing than other widely used prescriptions (e.g. Patterson et al., 1977; Hess et al., 1998)."*

Line 420: suggest using same color scale for Figure 11-B and Figure 8-B as for other panels in those two figures. It helps to compare the differences between EXT-HOM and INT-EXT.

We originally accepted the reviewer's suggestion of reversing the color mapping for SSA variations in the revised version of the manuscript. But once generated the new figures, we realized that our original color choice better represented the physics behind the variables. Lower values of SSA indicate more absorption by dust particles, which is typically associated with warming (and vice versa). To be consistent with the meaning of the colors for dust DRE (red = warming; blue = cooling), we decided therefore to maintain the original "reversed" color mapping for SSA variations: red = decreases = more absorption; blue = increases = more scattering. However, only for the map of absolute SSA (Figure 6-B), we replaced the blue with red, indicating again more absorption (lower SSA) with more intense red.

Line 454: do you mean "very large particles" between 30 micron and 32 micron in diameter? That doesn't seem to be a lot, even if the model's upper size limit is 2-micron larger than the assumption of AERONET.

We thank the reviewer for this and the following comments about this whole paragraph (lines 451 – 461 in the submitted version) that we totally rewrote (see next response).

Regarding this first comment, with "very large particles" we do not refer only to diameters between 30 and 32 µm. One of the assumptions in retrieving the size distribution in AERONET is that the volume size distribution must go to zero for diameters approaching to 30 µm. In principle, if particles near this size or larger are present in the column (more likely close to sources), and if those particles remain suspended for a time long enough to affect the measurements of radiance in AERONET, the above-mentioned retrieval assumption for the size distribution in AERONET could lead not only to missing diameters larger than 30 µm but also to underrepresenting slightly smaller diameters close to the threshold of 30 µm. This potential limitation of the AERONET inversion algorithm for

representing coarse particles is under discussion by researchers but its practical importance has not been quantified yet.

Lines 455-456: do you mean that AERONET over-predicts IRI but preserve the SSA? But, how does this relate to the model (HOM) underestimation of SSA from AERONET?

This comment follows from the previous one. The potential underrepresentation of coarse particles in AERONET, due to the cutoff at 30 μm in diameter for the particle volume distribution, would result in a larger retrieved imaginary index (compensating for the missing absorption by large particles) but would leave unaffected the retrieved SSA. This would happen because, while the AERONET SSA retrievals are heavily constrained by sky radiance measurements from the almucantar scan, the size distribution and imaginary index are co-retrieved. As mentioned in the previous comment, this potential issue in the AERONET size retrieval has not been quantified yet. Thus, we cannot assess to what extent our coarse dust mass distribution, that systematically represents higher fractions of coarse dust compared to AERONET (in the 8 – 32 μm diameter range; Figure S4 in the revised manuscript), contributes to our underestimation of AERONET SSA in the HOM case (Figure 12-A).

In the revised manuscript, we rephrased the entire paragraph about this issue in a clearer way, omitting some too detailed digressions and presenting it as an uncertainty (Section 4.3; Lines 614 – 625):

"*Finally, in addition to an excessively absorbing IRI from Sinyuk et al. (2003) (Section 3.2.1), our coarse size distribution may contribute to the general low bias of the HOM model SSA compared to AERONET (Figure 12-A), as larger particles have in general lower SSA. Despite our fine and coarse fractions of mass load are consistent with the semi-observational constraint by Adebiyi and Kok (2020), as discussed in Section 3.1.1, our model systematically represents higher fractions of coarse dust than indicated by AERONET in the selected dusty scenes (for diameters above 8μm according to Figure S4). However, while the AERONET SSA retrievals are heavily constrained by sky radiance measurements from the almucantar scan (e.g. Schuster et al., 2016), the AERONET inversion algorithm assumes that the volume size distribution falls to zero near diameters of 30μm (Dubovik et al., 2002). In principle, in addition to missing larger particles (e.g. Ryder et al., 2019), the assumed cutoff could also lead to underrepresenting smaller particles with diameters approaching to 30μm, if such coarse particles remain suspended long enough to be detected by the instrument. To date, this potential uncertainty in the AERONET size retrieval has not been quantified. Therefore, we cannot assess to what extent our underestimation of AERONET SSA in the HOM case results from excessive coarse particles in the model.*"

Additional References:

Ryder, C. L., Highwood, E. J., Walser, A., Seibert, P., Philipp, A. and Weinzierl, B. (2019). Coarse and giant particles are ubiquitous in Saharan dust export regions and are radiatively significant over the Sahara. *Atmos. Chem. Phys.*, 19, 15353 – 15376.

Line 460: what dust sources is the dust IRI from Sinyuk et al. (2003) representative of? It seems to overpredict the dust absorption (or IRI) in all the regions shown here from AERONET. It is a far stretch to apply one single value to the global scale, but at least it should be representative of the region where it is derived?

The reviewer rightly expects a stronger match between our model SSA and AERONET data in regions associated with the IRI used in our HOM case. Sinyuk et al. (2003) used observations at 4 locations to obtain the dust IRI, including Cape Verde and Dakar, both affected by Saharan dust

transported over the Atlantic Ocean, along with Bidibahn and Bondoukoui in Burkina Faso. According to our designated regions (Table 2, Figure 5), 3 out of these 4 stations fall within the Sahel, marked with down-triangle symbols in our SSA scatter plot (Figure 12). We do point out that some monthly means at Sahel locations in our control HOM case (Figure 12-A) are in agreement with AERONET values (for SSA ~ 0.92). However, it is also true that the model underestimates AERONET SSA in many other locations within the Sahel (see last response).

We added a new description of our underestimation of AERONET SSA in the HOM case that is shorter (because we moved the discussion about the AERONET size retrieval to the new Section 4.3) and incorporates the point of view suggested by the reviewer (Section 3.2.1; Lines 495 – 500):

"*The HOM model with globally homogeneous dust underestimates the AERONET SSA for most stations and months (Figure 12-A). This indicates that our IRI for unspeciated dust from Sinyuk et al. (2003) generally leads to excessive dust absorption compared to AERONET (Section 4.3) except for some stations within the Sahel, the region better represented by the IRI from Sinyuk et al. (2003), where model SSA around 0.92 closely align with AERONET values. Other exceptions to the low bias of model SSA, for some spring and summer months in North India, may be partly due to contamination of our dusty AERONET scenes by absorbing carbonaceous species (Go et al., 2022).*"

Line 486: is the low correlation between the three models and AERONET driven by under-represented temporal or spatial variability in the models?

This is an interesting question, but beyond the scope of this study. The low correlation could be due to a poor representation of the spatially varying source mineralogy, or to errors in transport that determines which source, in a certain moment, supplies the dust arriving at an AERONET station. It is difficult to distinguish the two types of error, because our model winds are calculated and therefore model transport differs from observed wind events. We would be able to attribute model errors (with respect to AERONET) to uncertainties in source composition or transport only in a statistical sense.

Line 509: AERONET DOD: do you mean AERONET AOD at the selected dusty sites? Or a fraction of the AERONET AOD for DOD? Please clarify.

We thank the reviewer for highlighting this detail that may be confusing for the reader. In this study, with AERONET DOD, we always refer to the AOD measured at the dusty scenes identified by our multiple filtering conditions on FVF, SSA and IRI (the same applies to AERONET SSA).

We added the following clarifying sentence to the revised manuscript (Section 2.3; Lines 319 – 320):

"*(In this work, with dust SSA and DOD from AERONET, we always refer to retrieved and measured values, respectively, in our sample of selected dusty scenes.)*"

Line 630: same question as above: why this IRI also leads to overestimated absorption in Sahara dust, from which it is derived.

As discussed in a previous comment, the observations used by Sinyuk et al. (2003) to obtain the IRI were taken at locations that are affected by Saharan dust but are mainly located within the Sahel region. Our comparison in Figure 12-A shows that some monthly means in the Sahel (as defined in our study) are in agreement with AERONET SSA. Nonetheless, this is not true for several other stations and months within the Sahel.

In addition to a potential contribution from our coarse size distribution to the general low bias of the model SSA compared to AERONET (mentioned in a previous response), in the revised manuscript we added some sentences about a potential uncertainty associated with our use of the IRI from Sinyuk et al. (2003) (Conclusions; Lines 678 – 683):

"*In contrast, assuming homogeneous composition according to the specific IRI from Sinyuk et al. (2003) results in excessive absorption with respect to AERONET in most stations and months, without reproducing the full variability of retrieved SSA. One potential uncertainty is that Sinyuk et al. (2003) retrieved dust IRI at only two UV wavelengths and inferred values from 0.331 to 0.670μm by log-linearly interpolating with AERONET IRI retrievals. We further extend these IRI values to encompass the entire UV-VIS band in our model. This extrapolation may not accurately capture IRI variations across VIS wavelengths, potentially overemphasizing absorption.*"